# Functional development of a V3/glycan-specific broadly neutralizing antibody isolated from a case of HIV superinfection

Mackenzie M Shipley[1†], Vidya Mangala Prasad[2†], Laura E Doepker[1], Adam Dingens[3], Duncan K Ralph[4], Elias Harkins[4], Amrit Dhar[4], Dana Arenz[1], Vrasha Chohan[1], Haidyn Weight[1], Kishor Mandaliya[5], Jesse D Bloom[3,6,7], Frederick A Matsen IV[4], Kelly K Lee[2]*, Julie M Overbaugh[1]*

[1]Division of Human Biology, Fred Hutchinson Cancer Research Center, Seattle, United States; [2]Department of Medicinal Chemistry, University of Washington, Seattle, United States; [3]Division of Basic Sciences, Fred Hutchinson Cancer Research Center, Seattle, United States; [4]Division of Public Health Sciences, Fred Hutchinson Cancer Research Center, Seattle, United States; [5]Coast Provincial General Hospital, Women's Health Project, Mombasa, Kenya; [6]Department of Genome Sciences, University of Washington, Seattle, United States; [7]Howard Hughes Medical Institute, Chevy Chase, United States

**Abstract** Stimulating broadly neutralizing antibodies (bnAbs) directly from germline remains a barrier for HIV vaccines. HIV superinfection elicits bnAbs more frequently than single infection, providing clues of how to elicit such responses. We used longitudinal antibody sequencing and structural studies to characterize bnAb development from a superinfection case. BnAb QA013.2 bound initial and superinfecting viral Env, despite its probable naive progenitor only recognizing the superinfecting strain, suggesting both viruses influenced this lineage. A 4.15 Å cryo-EM structure of QA013.2 bound to native-like trimer showed recognition of V3 signatures (N301/N332 and GDIR). QA013.2 relies less on CDRH3 and more on framework and CDRH1 for affinity and breadth compared to other V3/glycan-specific bnAbs. Antigenic profiling revealed that viral escape was achieved by changes in the structurally-defined epitope and by mutations in V1. These results highlight shared and novel properties of QA013.2 relative to other V3/glycan-specific bnAbs in the setting of sequential, diverse antigens.

*For correspondence:
kklee@uw.edu (KKL);
joverbau@fredhutch.org (JMO)

†These authors contributed equally to this work

## Introduction

Developing a vaccine that elicits broadly neutralizing antibodies (bnAbs) against HIV is considered critical to achieving protection from the extensive diversity of circulating HIV subtypes (*Burton and Hangartner, 2016*; *Doria-Rose, 2010*). The potency and protective efficacy of these bnAbs has been demonstrated in passive immunization trials (*Pegu et al., 2017*), however thus far, no vaccine design has been capable of eliciting potent HIV bnAbs. In part, this may be because HIV-specific bnAbs typically take years to develop in adults, requiring exposure of the immune system to significant, evolving viral diversity, and have unusual features including extended complementary determining regions (CDRs), widespread somatic hypermutation (SHM), and polyreactivity (*Mascola and Haynes, 2013*). Developing a vaccine that stimulates the production of neutralizing antibodies with such rare features may require a series of immunogens that each engage key intermediate members of the antibody's lineage, leading to eventual bnAb emergence (*Andrabi et al., 2018*; *Kwong and Mascola, 2018*). Using deep sequencing and phylogenetic approaches to define maturation pathways of bnAbs from their naive predecessors in the setting of natural infection provides insight that

can guide the design of tailored vaccine immunogens aimed at recapitulating the evolution and development of HIV-specific bnAbs (*Briney et al., 2016*; *Doria-Rose and Joyce, 2015*). Such antibody lineage reconstruction has been performed for a handful of HIV bnAbs, with particular focus on core bnAb epitopes including the CD4-binding site, the variable loop 1/2 (V1/V2) apex, and the conserved glycan supersite in V3 of HIV Envelope (Env) (*Bonsignori et al., 2017a*; *Doria-Rose et al., 2016*; *Garces et al., 2015*; *Kong et al., 2016*; *Krebs et al., 2019*; *IAVI Protocol C Investigators et al., 2017*; *IAVI Protocol C Investigators & The IAVI African HIV Research Network et al., 2016*; *Doria-Rose et al., 2014*; *Simonich et al., 2019*; *IAVI Protocol C Investigators et al., 2019*; *Wu et al., 2011*).

The glycan supersite in the V3 loop of Env is an advantageous target for vaccine immunogens because bnAbs recognizing this site are not germline restricted and often require less SHM than several other bnAb classes (*IAVI Protocol C Investigators & The IAVI African HIV Research Network et al., 2016*; *Simonich et al., 2019*). While V3/glycan-specific bnAbs have a shared requirement of the core glycan at site N332, they are also capable of recognizing heterogeneous glycan moieties with overlapping epitopes and they achieve this recognition using a variety of binding angles and approaches (*Barnes et al., 2018*; *Bonsignori et al., 2017a*; *Doores et al., 2015*; *Freund et al., 2017*; *Julien et al., 2013*; *Mouquet et al., 2012*; *Pejchal et al., 2011*; *Sok et al., 2016*; *Trkola et al., 1996*, p. 12; *Protocol G Principal Investigators et al., 2011*). The majority of HIV isolates are able to escape V3/glycan-specific bnAb recognition and neutralization by shifting the sites of glycosylation on the surface of Env (*Dingens et al., 2019*), but combination immunotherapy with bnAbs that target distinct epitopes has demonstrated the ability to thwart viral escape, without the emergence of resistance mutations (*Bar-On et al., 2018*; *Klein et al., 2012*; *Mendoza et al., 2018*). These findings support the use of genetically diverse immunogens during vaccination to generate a polyclonal neutralizing antibody response that targets distinct epitopes, as is observed following natural HIV superinfection (*Cortez et al., 2015*; *Doria-Rose and Joyce, 2015*; *Williams et al., 2018*).

HIV superinfection (SI) represents a unique setting to study the evolution of bnAbs, as antigenic stimulation with two genetically distinct virus strains may lead to the development of neutralizing antibody responses through mechanisms that differ from that of single infection. Indeed, it has been shown that plasma from SI individuals is more broad and potently neutralizing than plasma obtained from time-matched samples of single HIV infection (*Cortez et al., 2012*; *Powell et al., 2010*). Other studies of SI have found an additive effect to the neutralizing antibody response following secondary infection, in which disparate antibody responses are generated to distinct viral epitopes (*Sheward et al., 2018*), suggesting that superinfection may drive a polyclonal neutralizing antibody response as opposed to only a boost in memory B cell responses. This idea is supported by plasma mapping data demonstrating the lack of a dominant viral epitope signature recognized by neutralizing antibodies present in plasma from 21 SI women (*Cortez et al., 2015*). These data suggest that immune responses to superinfection are distinct from those to single HIV infection and, thus, could provide additional clues as to how diverse and protective bnAbs evolve. Moreover, immune stimulation by two distinct viruses produces a situation that emulates aspects of sequential vaccination with different antigenic variants. This scenario allows us to delineate how antibody responses evolve following sequential exposure to distinct heterologous antigens. To date, examination of bnAb evolution and functional development has only been assessed for one case of HIV superinfection, that of participant CAP256, who developed a potent V1/V2-specific bnAb lineage (CAP256-VRC26) that recognized the SI virus (*Doria-Rose et al., 2016*; *Doria-Rose et al., 2014*).

Here, we detail the functional development of a V3/glycan-specific bnAb QA013.2 isolated approximately 6 years post initial infection from a HIV superinfected woman in Mombasa, Kenya (*Williams et al., 2018*). This bnAb binds the V3 glycan at site N332, with a footprint similar to that of the bnAb PGT128 (*Williams et al., 2018*). SHM gave QA013.2 the ability to bind and recognize the initial infecting virus that lacks the N332 glycan, despite the bnAb evolving to only neutralize the SI virus (*Williams et al., 2018*). A cryo-EM structure of the mature Fab bound to heterologous HIV Env trimer reveals that QA013.2 evolved to establish a conformational epitope that spans V3, with the majority of the antigen contacts mediated by the antibody's heavy chain, with additional contributions from residues in the light chain framework (FWR) and CDR loops. However, unlike many other V3/glycan-specific bnAbs reported in the literature, this bnAb emerging from HIV superinfection relies on residues that span FWRH1 and CDRH1 to achieve cross-clade Tier two heterologous neutralization, while residues in CDRH3 are less impactful. The heavy chain-driven antigen recognition

by QA013.2 elicited viral escape at sites beyond the V3 region of Env in the variable loop 1, suggesting a more complex conformational epitope than other V3/glycan-specific bnAbs. These findings shed light on at least one demonstrable route to an immune response achieving bnAb breadth – a prerequisite for vaccine immunogen design.

## Results

### QA013.2 clonal lineage development

Study participant QA013 was initially infected with a HIV clade D virus in 1995. At just over one year post initial infection (pii), a clade A variant was identified that had not been detected at the previous visit a few months earlier, indicating superinfection (SI) occurred in the intervening period: approximately 264–385 dpii (*Chohan et al., 2005*). Both viruses persisted in this individual for at least 6 years, and our lab previously isolated the bnAb QA013.2 at 2282 dpii (just over 6 years pii), along with several less potent nAbs that also contributed to plasma breadth. (*Williams et al., 2018*). To investigate the functional development of this V3/glycan-specific bnAb, we performed deep sequencing of the subject's antibody repertoire using four longitudinal peripheral blood mononuclear cell (PBMC) samples spanning time points from pre-HIV infection (D-379) to 765 dpii (D765) (*Figure 1A*). With these deep sequencing data, computational inference methods designed for antibody lineage analyses were used to infer the probable naive B cell receptors (BCRs) that seeded the heavy and light clonal families of QA013.2. We then performed ancestral sequence reconstruction using Bayesian phylogenetics to infer the most probable routes of development from naïve BCR sequence to mature sequence (*Figure 1B*, *Figure 1—figure supplements 1–4*; *Dhar et al., 2020*; *Ralph and Matsen, 2016*). Antibodies were constructed based on the most statistically probable routes of development for the QA013.2 bnAb heavy and light chains, which involved eight variable heavy chain (VH) intermediate sequences and six variable light chain (VL) intermediate sequences between respective naïve BCR and mature sequences (*Figure 1C*, *Supplementary file 1)* Of note, several of the inferred intermediate sequences shared $\geq$98% nucleotide identity with sequences sampled from the longitudinal B cell repertoire of QA013, and in some cases, there were identical sequences observed (*Figure 1C*, *Supplementary file 2*).

The heavy chain clonal family of QA013.2 included a total of 42 members and this clonal family ranked among the top 10 largest clonal family clusters that were sampled in this subject's immune repertoire. These clonally related sequences were derived from an inferred naïve BCR that used VH3-7*01, VD1-1*01, and VJ5*02 genes and had a 21-residue CDRH3 loop. The VH lineage matured over the course of six years to accumulate 20% somatic hypermutation (SHM) – which corresponded to a total of 38 amino acid mutations across the variable region in the mature QA013.2 bnAb (*Figure 1C*). The QA013.2 light chain clonal family included >1000 members likely due to overclustering, but it did not rank among the top 10 largest light chain clonal families that were sampled in the repertoire (also due to overclustering, see Materials and methods). The inferred naive light chain BCR used germline genes VL1-40*01 and VJ2*01 and the lineage acquired fewer mutations over the course of maturation, with a total of 12% SHM present in the mature lambda light chain (*Figure 1C*).

From the deep sequencing data, which was informed by samples up to 765 dpii, we observed that the inferred antibody lineages acquired limited SHM between the inferred naive BCRs and the latest inferred intermediates (Int6$_{VH}$ and Int5$_{VL}$). More than 70% of the total SHM (84 of 117 nucleotide mutations across VH and VL regions combined) present in QA013.2 was not detected in the sequence data through 765 dpii, suggesting that either we did not sample sequences containing these mutations, or that these mutations accumulated between 765 dpii and the time when QA103.2 was isolated at 2282 dpii. Thus, we can only reliably infer the lineage through the early stages of development; the later stages of the bnAb's evolutionary pathway are beyond the scope of the computationally inferred lineage (*Figure 1B,C*). Despite this lack of resolution, the lineage and longitudinal development that we were able to decipher offer a means to dissect the determinants of antibody binding, potency, and breadth for QA013.2. Here, we defined neutralization breadth as the mediation of cross-clade heterologous neutralization of Tier 2 HIV pseudoviruses as measured by the TZM-bl assay.

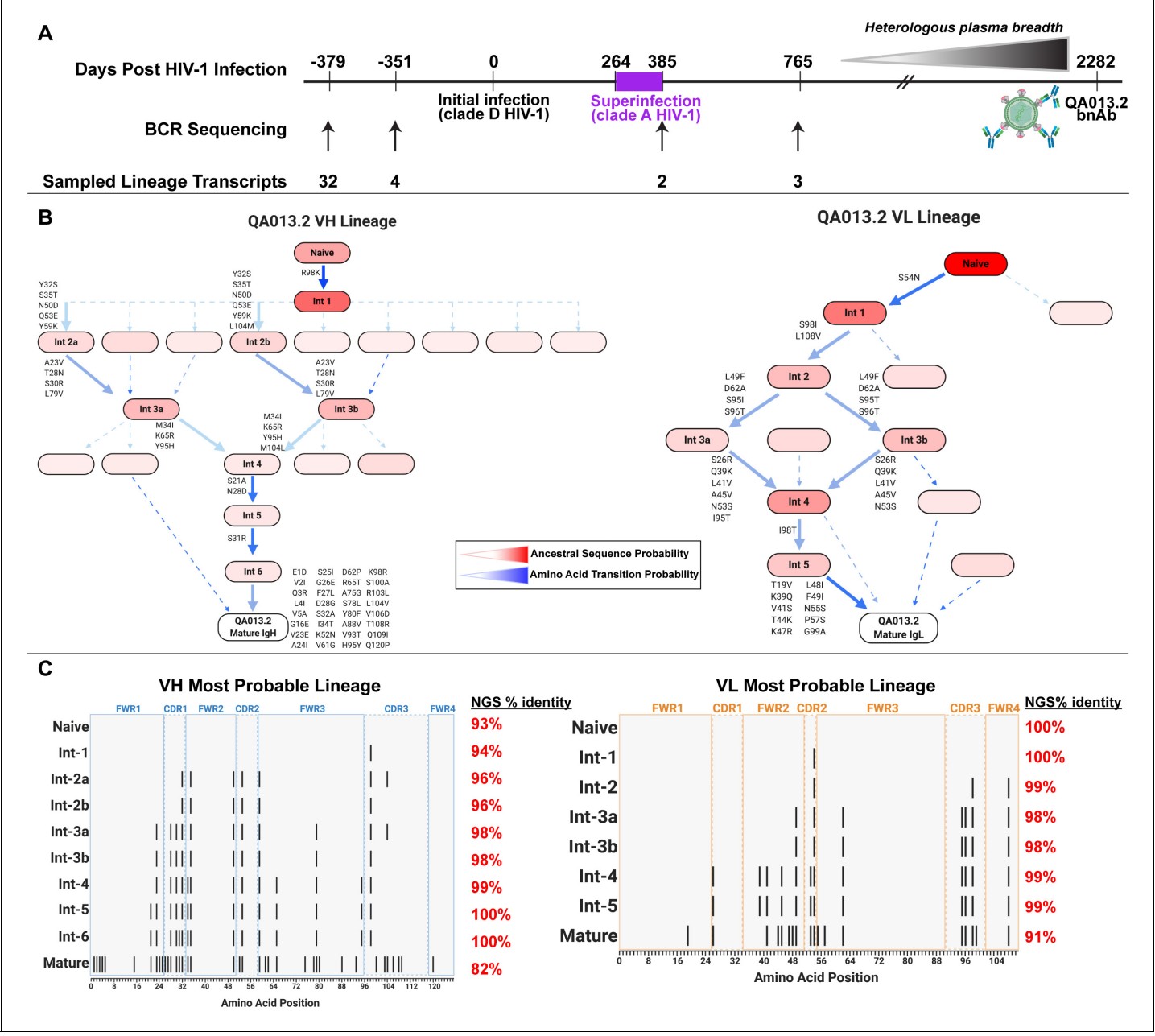

**Figure 1.** QA013.2 inferred clonal lineage development. (**A**) Timeline of HIV infection, isolation of QA013.2 bnAb, and longitudinal PBMC samples used for antibody variable region sequencing. Days post initial HIV infection are listed above the timeline and black arrows below the timeline correspond to the specific PBMC samples that were used for deep sequencing. Approximate heterologous plasma breadth as determined previously (*Cortez et al., 2012*) is depicted above the timeline, which is not to scale. Estimated timing of superinfection as previously described (*Chohan et al., 2005*) is shown in purple on the timeline, with the timing of superinfection estimated as the midpoint, 324.5 days post initial infection. The QA013.2 bnAb was not isolated until 2282 days, or about 6.2 years post initial infection, as indicated. (**B**) Lineage graphics representing probable developmental paths for heavy and light chain lineages between inferred naïve BCR and mature QA013.2 sequences. Nodes represent inferred ancestral sequences, while arrows in between each node represent amino acid transitions. The red shading of nodes is proportional to the posterior probability that this ancestral sequence was present in the true lineage. For each given node, the blue shading of arrows arising from that node is proportional to the corresponding transition probability. Low probability nodes were filtered out, resulting in some incomplete pathways within the graphics. Dashed arrows indicate possible developmental paths that were not chosen based on low probability. See *Figure 1—figure supplements 1–4*. (**C**) Probable lineage sequences for the developing QA013.2 heavy (blue) and light chains (orange) are displayed in their inferred chronological order. With respect to the inferred naive sequence at the top of each lineage, variable region amino acid substitutions are indicated by black lines. Dashed lines demarcate the CDRs that are flanked by FWRs. Red percentages to the right of each sequence represent the nucleotide identity of each computationally inferred lineage member to sampled NGS sequences present in the longitudinal B cell repertoire of QA013, rounded to the nearest percent. See also

*Figure 1 continued on next page*

*Figure 1 continued*

*Supplementary files 1* and *2* for fasta files of computationally inferred lineage members and sampled NGS sequences with high nucleotide identity to inferred lineage members.

The online version of this article includes the following figure supplement(s) for figure 1:

**Figure supplement 1.** Phylogenetic relationship of sampled QA013.2 heavy and light chain clonal family sequences from four longitudinal PBMC samples.

**Figure supplement 2.** Linearham lineage trajectory graphics of QA013.2 VH inferred lineage members from inferred naive antibody sequence to mature.

**Figure supplement 3.** Linearham lineage trajectory graphics of QA013.2 VL inferred lineage members from inferred naive antibody sequence to mature.

**Figure supplement 4.** Partis clonal family cluster for QA013.2 clonal family containing VH sequences sampled from QA013 over four longitudinal timepoints.

## QA013.2 heavy chain maturation enables antigen binding and neutralization

To determine whether key residues mediating breadth were represented in the inferred intermediate sequences defined through 765 dpii, we selected these intermediates (Int6$_{VH}$ and Int5$_{VL}$; *Figure 1*), as well as inferred naive heavy and light chain BCR sequences and paired them together in various combinations (*Figure 2A*) to test their capacity to bind and neutralize HIV relative to the mature QA013.2 bnAb. The resulting monoclonal antibodies (mAbs) were tested for binding to autologous and heterologous Envelope protein (Env) by biolayer interferometry (BLI). The inferred naive BCR had weak but detectable binding to the Env-gp120 from a clade A autologous superinfecting virus variant from 765 dpii; there was even weaker binding to heterologous clade A BG505. SOSIP.664 trimer (*Figure 2B,C*). Although these data were insufficient for calculating reliable binding kinetics, they suggest that the naïve BCR of the QA013.2 bnAb lineage is capable of recognizing the superinfecting virus. The inferred naive BCR did not detectably bind to the clade D virus from initial infection (*Figure 2—figure supplement 1*).

By contrast, the mature QA013.2 bnAb exhibited strong binding kinetics to both the autologous SI gp120 and heterologous BG505 Env trimer, with equilibrium dissociation constants of 94.9 nM and 3.2 nM, respectively. A mAb consisting of the mature heavy chain paired with the inferred naive light chain (mat$_{VH}$ 0$_{VL}$) bound to autologous SI gp120; however, the binding affinity was much weaker relative to the mature bnAb and not suitable for calculating binding kinetics. This same mAb bound to heterologous BG505 Env trimer (K$_D$ = 89.6 nM) with an affinity that was approximately 30-fold less than that of the mature bnAb (*Figure 2B,C*). The converse mAb pairing (0$_{VH}$ mat$_{VL}$) did not result in any binding capacity to either tested antigen.

Intermediate mAb Int6$_{VH}$ Int5$_{VL}$, with 6% total SHM, represents the final inferred intermediates from the deep sequencing data, and they were detected in the sampled NGS sequences at 99–100% nucleotide identity, further validating the sequence of these inferred intermediates (*Supplementary file 2*). We could not detect binding of the intermediate mAb to the gp120 of the autologous SI virus, but there was clear evidence of binding to heterologous BG505 Env trimer (K$_D$ = 513 nM) (*Figure 2C*), demonstrating HIV Env specificity. However, the binding affinity to heterologous trimer was more than 100-fold lower when compared to the mature bnAb. Together these studies demonstrate that VH is important for facilitating initial HIV recognition and binding. This finding is supported by data collected for both autologous Env-gp120 and heterologous native-like trimer, despite the native-like Env trimer yielding tighter binding affinities and dissociation constants for all tested mAbs. While early affinity maturation of the QA013.2 VH lineage led to measurable binding to heterologous Env antigen (*Figure 2C*), higher affinity binding required subsequent mutations, including acquisition of 30 non-synonymous mutations across the variable region between Int6$_{VH}$ and mature$_{VH}$. (*Figures 1B,C* and *2B,C*).

We tested the same lineage mAbs for neutralization using a subset of HIV Env pseudoviruses that were neutralized by the mature QA013.2 bnAb, including the same clade A autologous Env variant from 765 dpii used to generate gp120 for binding studies (*Williams et al., 2018*). Neutralization was not observed for the inferred naive BCR (0$_{VH}$0$_{VL}$) (*Figure 2D*). While binding was detected between the intermediate mAb (Int6$_{VH}$Int5$_{VL}$) and the heterologous BG505 Env trimer, it did not neutralize the same virus (*Figure 2C,D*). Thus, the inferred naive and intermediate Int6$_{VH}$Int5$_{VL}$ mAbs represent

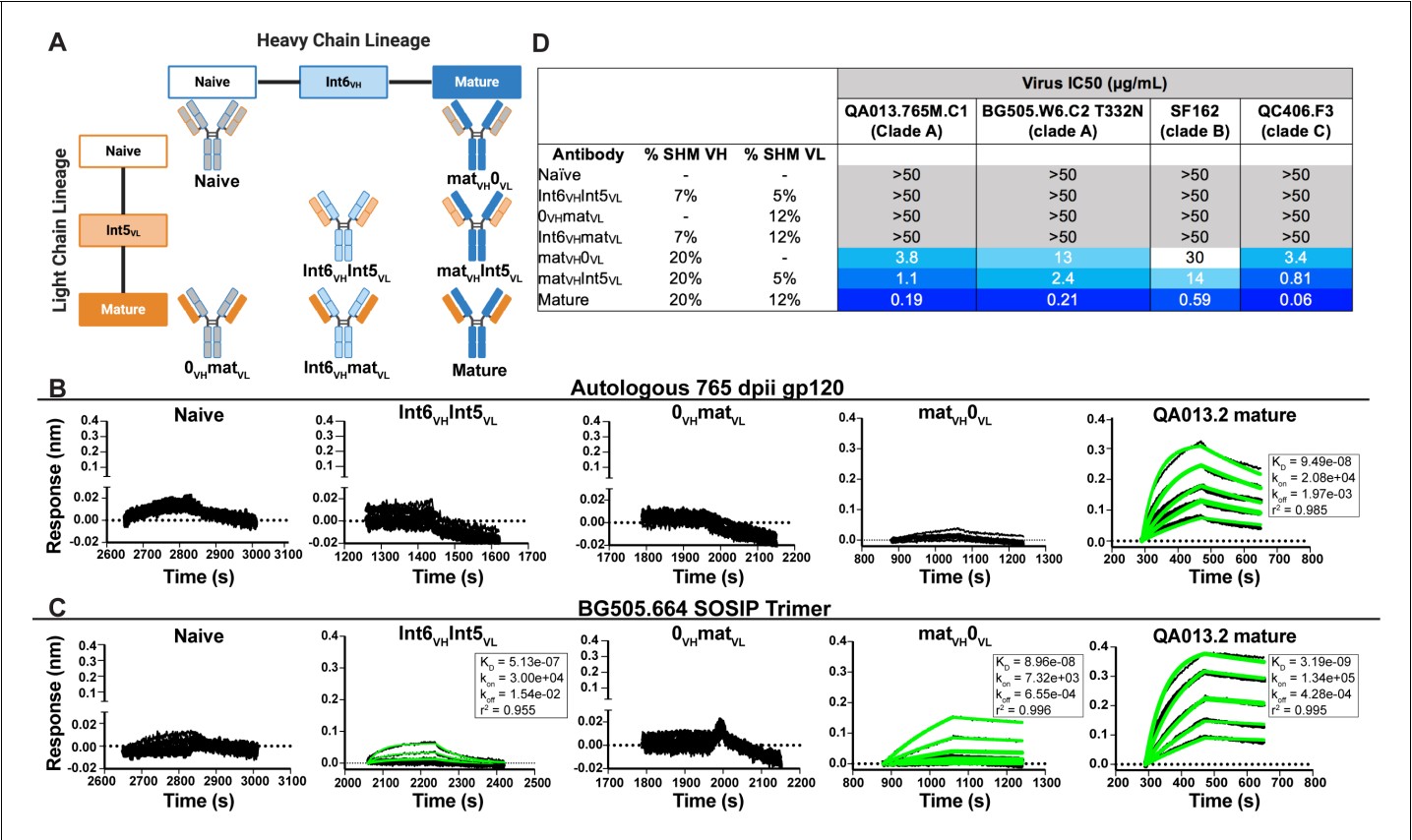

**Figure 2.** Binding and neutralization analyses of QA013.2 inferred VH and VL antibody chains demonstrates that heavy chain maturation is critical for bnAb function. (A) Schematic depicting antibody chain pairings of inferred lineage member heavy (blue) and light chains (orange) with one another. (B, C) Biolayer interferometry (BLI) analysis of QA013.2 inferred lineage mAbs at 10 μg mL$^{-1}$ binding to (B) the Env-gp120 monomer from a clade A autologous superinfecting virus variant from 765 dpii (QA013.765M.C1, 2 μM) and (C) the heterologous clade A Env trimer (BG505.SOSIP.664, 500 nM). All BLI curves were subjected to double reference background subtraction and data were fit to a global model of 1:1 ligand:analyte binding to obtain $K_D$, $k_{on}$, and $k_{dis}$ values. The resulting lines of best fit are shown in green. Dotted line across the graphs marks 0.00 on the y-axis. All data are representative of at least two independent experiments. (D) QA013.2 inferred lineage mAb neutralization of autologous and heterologous pseudoviruses as measured by the TZM-bl assay. The half maximal inhibitory concentration ($IC_{50}$) for each tested pseudovirus is shown in a row across the top of the table. Darker blue indicates more potent neutralization, while white demonstrates weak neutralization, and grey represents no neutralization observed at the highest antibody concentration tested. Simian immunodeficiency virus (SIV) was used as a negative control; none of the tested mAbs showed evidence of SIV neutralization (data not shown). $IC_{50}$ values are the average of at least two independent replicates. See *Figure 2—figure supplement 1*, *Figure 2—source data 2*, *Figure 2—source data 3* and *Figure 2—source data 1*.

The online version of this article includes the following source data and figure supplement(s) for figure 2:

**Source data 1.** Biolayer Interferometry Source Data 1.

**Source data 2.** Biolayer Interferometry Source Data 2.

**Source data 3.** Neutralization Source Data.

**Figure supplement 1.** QA013.2 inferred naïve BCR binding to initial infecting virus Env-gp120 monomer.

**Figure supplement 1—source data 1.** Biolayer Interferometry Source Data 1.

**Figure supplement 1—source data 2.** Biolayer Interferometry Source Data 2.

early members of the lineage with the capacity to bind HIV Env trimer that have not yet acquired the ability to neutralize either autologous or heterologous variants. While the inferred Int6$_{VH}$Int5$_{VL}$ intermediate was validated by near-perfect sequence matches in the sampled NGS sequences (*Supplementary file 2*), we cannot rule out the possibility that another lineage member with neutralization capacity was also present by 765 dpii, because deep sequencing antibody variable regions from blood PBMC samples often results in incomplete sampling (*Horns et al., 2019*;

**Table 1.** Sequencing statistics of longitudinal PBMC samples from subject QA013.

| PBMC time point | Live PBMC count | PBMC viability | Ab chain | Raw MiSeq reads | Productive deduplicated sequences | Sequence coverage of sampled blood PBMCs | Sequence coverage of QA013 D2282 whole-body blood* repertoire |
|---|---|---|---|---|---|---|---|
| D-379 | 2.60E+06 | 96% | IgM | 1358426 | 2700 | 0.8% | 0.002% |
| | | | IgG | 1357742 | 8350 | 14% | 0.03% |
| | | | IgK | 1701268 | 21043 | 6% | 0.01% |
| | | | IgL | 1528383 | 27458 | 7% | 0.02% |
| D-351 | 2.50E+06 | 100% | IgM | 1186865 | 2213 | 0.7% | 0.002% |
| | | | IgG | 1290351 | 7093 | 12% | 0.03% |
| | | | IgK | 1724651 | 19835 | 5% | 0.01% |
| | | | IgL | 1116583 | 23489 | 6% | 0.01% |
| D385 | 3.90E+06 | 100% | IgM | 1299494 | 2039 | 0.4% | 0.001% |
| | | | IgG | 1512588 | 6222 | 7% | 0.02% |
| | | | IgK | 2058137 | 22252 | 4% | 0.01% |
| | | | IgL | 1454466 | 27107 | 5% | 0.01% |
| D765 | 6.80E+06 | 96% | IgM | 1264049 | 2480 | 0.3% | 0.001% |
| | | | IgG | 1465034 | 7424 | 12% | 0.03% |
| | | | IgK | 2083938 | 21738 | 2% | 0.00% |
| | | | IgL | 1556226 | 26530 | 3% | 0.01% |

*Assuming PBMC samples were from a 10 mL blood draw and that adults have 4500 mL total blood volume.

*Laserson et al., 2014*). Indeed, we observed low sequence read depth across all QA013 PBMC sample timepoints and antibody isotypes (*Table 1*).

When the mature VH was paired with the inferred naive light chain ($mat_{VH}0_{VL}$), autologous and cross-clade Tier two heterologous neutralization was achieved, suggesting that the approximate timeframe in which neutralization-mediating mutations were acquired in VH likely occurred between 765 dpii (resolution of computationally inferred lineage) and 2282 dpii (isolation of mature bnAb QA013.2). The inferred naive heavy chain paired with the mature light chain ($0_{VH}mat_{VL}$) did not result in any detectable neutralization of the virus panel (*Figure 2D*), suggesting that affinity maturation of the heavy chain is critical for imparting both binding and neutralization. When $Int5_{VL}$ was paired with the mature heavy chain ($mat_{VH}$ $Int5_{VL}$), there was an average fourfold increase in neutralization potency for all viruses tested compared to $mat_{VH}0_{VL}$ (*Figure 2D*).

### Cryo-EM structure of QA013.2 Fab bound to Env trimer demonstrates widespread antibody contacts and shared structural characteristics with other V3/glycan-specific bnAbs

To better identify which residues contribute to QA013.2's ability to bind and neutralize Env, we performed single particle cryo-electron microscopy (cryo-EM) on the QA013.2 Fab and BG505. SOSIP.664 trimer complex (*Figure 3A,B*). The known structure of BG505.SOSIP.664 trimer (PDB ID: 5aco) was fit into the resulting 4.15 Å resolution EM density map along with a homology model for the variable domain of QA013.2 Fab (*Figure 3A,B*). The map density corresponding to the Fab variable domain was well resolved, with prominent loops and separation of strands in beta sheet regions. These features facilitated high confidence docking of the Fab homology model into the EM map. Subsequent real-space refinement of the BG505.SOSIP.664 + QA013.2 Fab model against the map enabled a good fit of secondary structural elements within the EM density (*Figure 3—source data 1*). The complex structure of QA013.2 Fab bound to BG505.SOSIP.664 delineated antigen contacts characteristic of other V3/glycan-specific bnAbs, including conserved glycans as well as the linear GDIR motif (*Barnes et al., 2018*; *Mouquet et al., 2012*; *Pejchal et al., 2011*).

From the cryo-EM structure, it is apparent that QA013.2 interacts not only with the N332 glycan as previously reported (*Williams et al., 2018*), but also with the glycan at site N301, and the linear

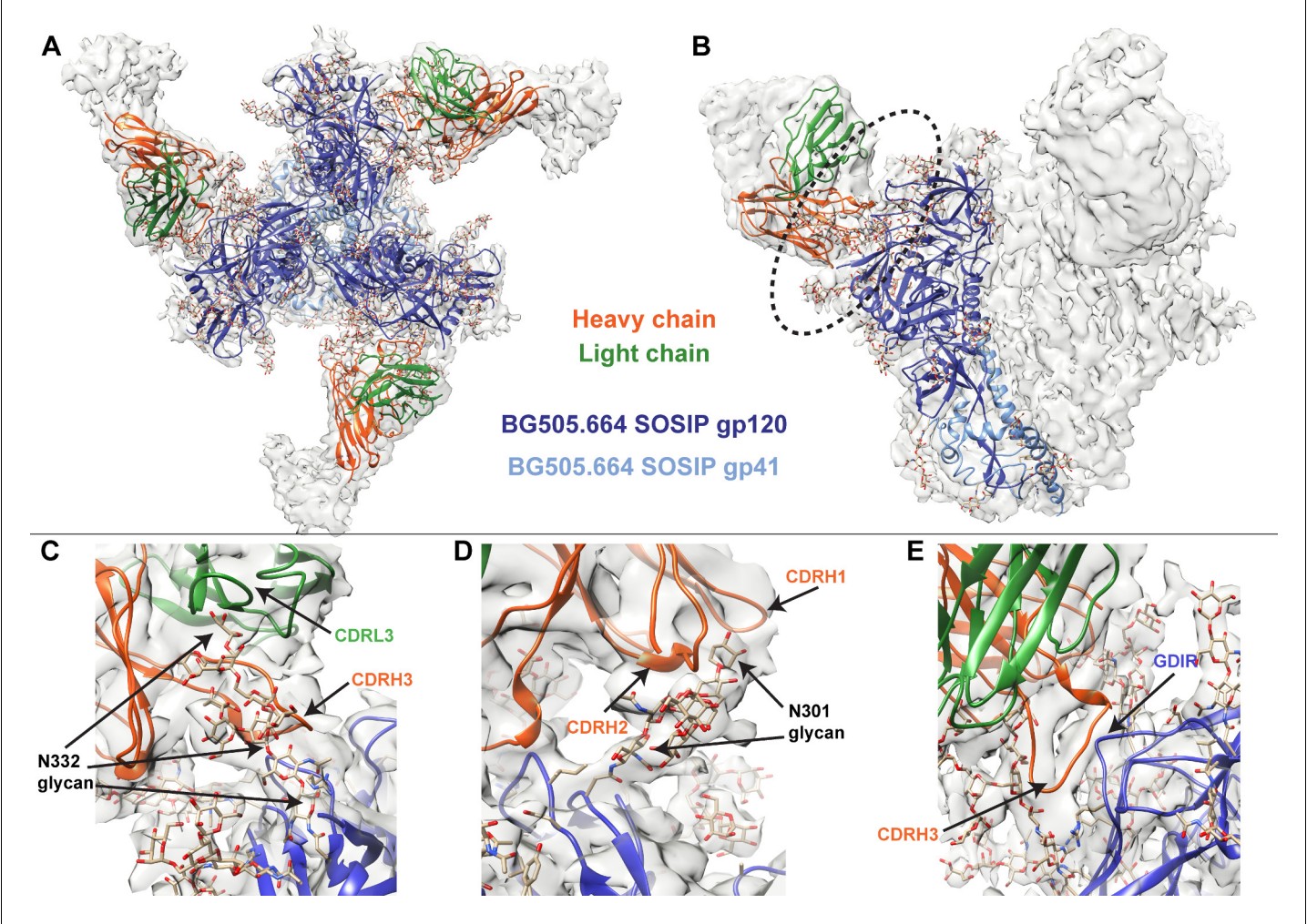

**Figure 3.** Cryo-EM structure of QA013.2 Fab bound to BG505.SOSIP.664 trimer.  (A) Cryo-EM reconstruction of QA013.2 Fab bound to BG505. SOSIP.664 trimer. Gp120 chains are shown in dark blue, gp41 in light blue, Fab heavy chain in orange, and Fab light chain in green. (B) Side-view of cryo-EM reconstruction with a single copy of the PDB model shown. The black oval indicates the region of interaction between QA013.2 Fab and HIV BG505 gp120 that is enlarged and highlighted in detail in panels C-E. (C) N332 glycan interaction with CDRL3 and CDRH3. (D) N301 glycan interaction with CDRH1 and CDRH2. (E) Heavy chain CDR3 interacts with the linear GDIR motif of gp120. See also *Figure 3—figure supplements 1* and *2* and *Figure 3—source data 1*.

The online version of this article includes the following source data and figure supplement(s) for figure 3:

**Source data 1.** BG505.SOSIP + QA013.2 Fab cryo-EM data collection, refinement parameters, and model statistics.
**Source data 2.** BG505.SOSIP + QA013.2 Fab cryo-EM map and coordinates.
**Figure supplement 1.** Cryo-EM features of QA013.2 Fab heavy chain.
**Figure supplement 2.** Fourier Shell Correlation curves.

GDIR motif at the base of the V3 loop (*Figure 3C–E*). Key regions of the antibody paratope include the CDRH1 and CDRH2 which are nearest the N301 glycan, as well as CDRH3 and CDRL3, which are in close proximity to the N332 glycan. We observe that the CDR loops of the heavy chain contain many polar and charged residues, which likely facilitate hydrogen bonding interaction with the Env glycans and the charged residues in the GDIR motif (*Figure 3—figure supplement 1A*). The CDRH3 loop is distinctively long (21 residues) and appears to tilt away from the rest of the heavy chain toward the light chain, generating a cleft between CDRH2 and CDRH3 and exposing an extended interface that has been observed for other V3/glycan-specific bnAbs including 10–1074 and PGT121 (*Figure 3—figure supplement 1B*; *Mouquet et al., 2012*). Seven key residues in the Fab CDRH3 loop (L103 – I109) lie adjacent to the conserved GDIR motif. Based on our sequencing data of the

sampled repertoire of this subject, these amino acids (excluding Y105 and D107) were likely introduced during affinity maturation of the heavy chain between 765 dpii and 2282 dpii because they were not detected by NGS through 765 dpii (*Figure 1C*), suggesting a potential role for these residues in facilitating antibody breadth and/or potency.

## Substitutions in FWRH1-CDRH1 are required for bnAb neutralization breadth

With paratope data in hand, we had the opportunity to show which regions of QA013.2 were playing a role in its functional activity by dissecting the regional and individual residue contributions interpreted through the structure. We began by examining the functional importance of VH mutations surrounding the CDR loops that were present in the mature antibody but not in the ≤765 dpii

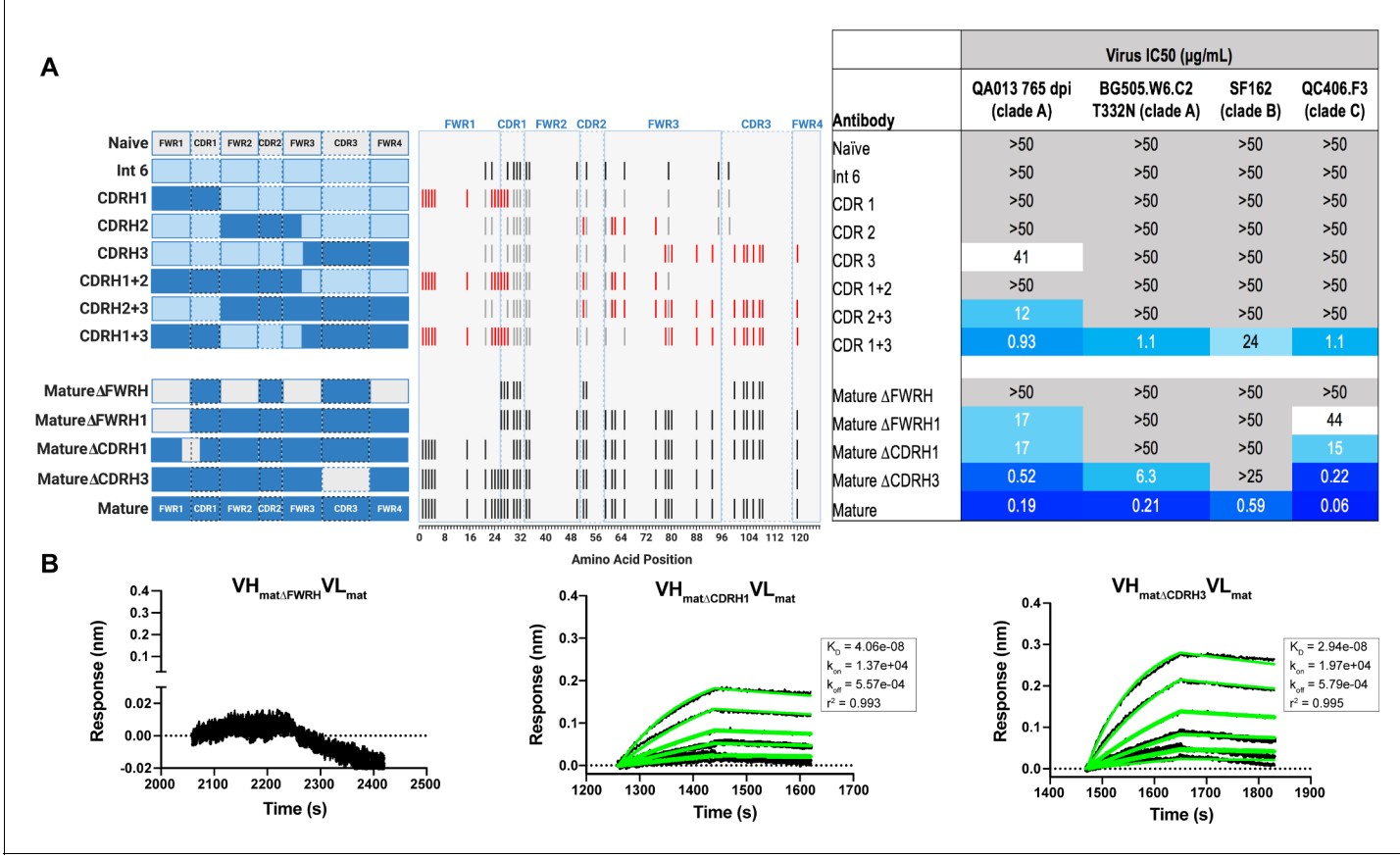

**Figure 4.** Interrogating the QA013.2 heavy chain paratope through binding and neutralization studies using chimeric antibodies. (**A**) Schematic of heavy chain chimeras that were generated and paired with the mature light chain. Names of each heavy chain variant are listed on the left, with colored bars representing the VH region including FWR and CDR loops colored by lineage member to visualize how each variant was generated. Inferred naïve heavy chain is represented in grey, latest inferred heavy chain intermediate (Int6$_{VH}$) is shown in light blue, and the mature heavy chain is shown in dark blue. The middle plot shows the individual mutations represented as black lines across VH relative to the inferred naïve BCR, with CDRs demarcated by dashed lines. Red lines represent all mutations *added* to the Int6$_{VH}$ template. Neutralization table on the far right shows the average IC$_{50}$ values for each tested chimeric antibody across a panel of pseudoviruses shown at the top. Characteristics of the neutralization table are consistent with those listed in the *Figure 2* legend. (**B**) Biolayer interferometry kinetic curves of a select subset of tested antibody variants (10 µg mL$^{-1}$ against heterologous trimer - BG505.SOSIP.664 at 500 nM). BLI curves were analyzed as described in the *Figure 2* legend. See *Figure 4—source data 1*, *Figure 4—source data 2* and *Figure 4—source data 3*. See also *Supplementary file 3* for fasta file of all chimeric and mutant VH and VL antibody chains used in *Figures 4*, *5*, *6*.

The online version of this article includes the following source data for figure 4:

**Source data 1.** Biolayer Interferometry Source Data 1.
**Source data 2.** Biolayer Interferometry Source Data 2.
**Source data 3.** Neutralization Source Data.

inferred intermediates. We synthesized six additional variations of QA013.2 VH, illustrated in *Figure 4A*, that we based on Int6$_{VH}$ as the template and included all CDR-localized and CDR-adjacent FWR residues present in the mature QA013.2 VH (*Supplementary file 3*). All VH chimeras were paired with the mature light chain (mat$_{VL}$) and tested for neutralization capacity (*Figure 4A*). Mutations in individual CDRs, including adjacent FWR mutations, were not sufficient to achieve potent neutralization comparable to the mature antibody when paired with the mature light chain. We did observe weak autologous neutralization when mutations surrounding FWRH3-CDRH3 were added to Int6$_{VH}$. Mutations spanning both FWRH1-CDRH1 and FWRH3-CDRH3 of the mature VH together were required to obtain heterologous neutralization, consistent with the cryo-EM map of key antigen contacts (*Figures 3* and *4A*). Interestingly, the determinants of autologous and heterologous neutralization in this antibody lineage were different, in which autologous virus neutralization required less SHM and was facilitated by residues spanning CDRH2-FWRH3-CDRH3 (*Figure 4A*). To determine the importance of the framework residues and examine loss of function, we reverted all framework mutations in the mature VH back to the inferred naive BCR residues (ΔFWR). This reversion resulted in no detectable binding or neutralization activity, demonstrating that residues in the heavy chain FWRs are essential for QA013.2 binding and neutralization, as has been noted for other HIV bnAbs (*Figure 4A,B*; *Klein et al., 2013*).

In a complementary approach, we used the mature VH as a template and reverted all of the mutations in FWRH1 alone (ΔFWRH1), as well as those that spanned FWRH1-CDRH1 (ΔCDR1$_{VH}$) and CDRH3 (ΔCDR3$_{VH}$) back to the inferred naïve VH and as above, paired these variants with the mature light chain (*Figure 4A*). We observed identical increases in the half maximal inhibitory concentration (IC$_{50}$) neutralization of autologous superinfecting virus for both the ΔFWRH1 and the ΔCDRH1 chimeric antibodies, with a 89-fold increase in IC$_{50}$ neutralization relative to the mature bnAb (*Figure 4A*). We also observed a dramatic reduction in heterologous virus neutralization for these same VH chimeric antibodies (ΔFWRH1 or ΔCDRH1). Specifically, we observed 733-fold and 250-fold increases in the IC$_{50}$ for ΔFWRH1 and ΔCDRH1 chimeric antibodies respectively, corresponding to decreased neutralization of clade C virus QC406.F3 (*Figure 4A*). Reversion of the six residues spanning FWRH1-CDRH1 also led to weakened binding to heterologous BG505.SOSIP.664 trimer (K$_D$ = 40.6 nM) relative to the mature bnAb (*Figure 4A,B*). When the CDRH3 was reverted back to naïve (ΔCDR3$_{VH}$), binding and neutralization of both autologous and heterologous virus was maintained, but there was an impact on potency across all viruses tested; although this antibody could still neutralize a heterologous clade C variant, it did not neutralize SF162 at the highest antibody concentration tested (*Figures 3* and *4A,B*). These data suggest that the CDRH3 does contribute to the overall breadth and potency of QA013.2, while mutations in FWRH1-CDRH1 are critical for achieving complete cross-clade Tier two neutralization breadth.

To further interrogate the above hypothesis, we generated single point mutations in the mature VH at residues of interest in FWRH1-CDRH1 and CDRH3 that reverted them to naïve (*Figure 5A,B*). These heavy chain point mutants were also paired with the mature light chain and tested for binding to Env and neutralization. We found that four of the six residues spanning FWRH1-CDRH1 individually contributed to the mature bnAb's ability to neutralize autologous and heterologous virus. Residues E26, G28, and I24 were most critical, with average increase in IC$_{50}$ of 191-, 27-, and 25-fold, respectively, across all pseudoviruses tested when these residues were reverted to naïve (*Figure 5C, D*). While reversion of individual CDRH3 residues did not abolish neutralization, which correlates with the chimeric antibody results (*Figure 4*), one residue in particular (D106) demonstrated a much higher IC$_{50}$ value when mutated back to naive. D106 is adjacent to the GDIR motif in the structure. Reversion of this residue to naive did not inhibit clade A autologous virus neutralization, but did substantially impact heterologous neutralization, with an average 40-fold increase in IC$_{50}$ across all tested pseudoviruses (*Figure 5E*). However, the D106V point mutant only resulted in a modest decrease in binding to heterologous BG505.SOSIP.664 trimer relative to the mature bnAb (*Figure 5F*). These data suggest that while D106 is not critical for Env binding, this residue does contribute to heterologous neutralization, likely by interacting directly with the linear GDIR motif. Taken together, it is clear that although the antibody paratope of QA013.2 includes the CDRH3, where interaction with the N332 glycan and GDIR motif occurs, the QA013.2 paratope is also comprised of critical residues in FWRH1 and CDRH1.

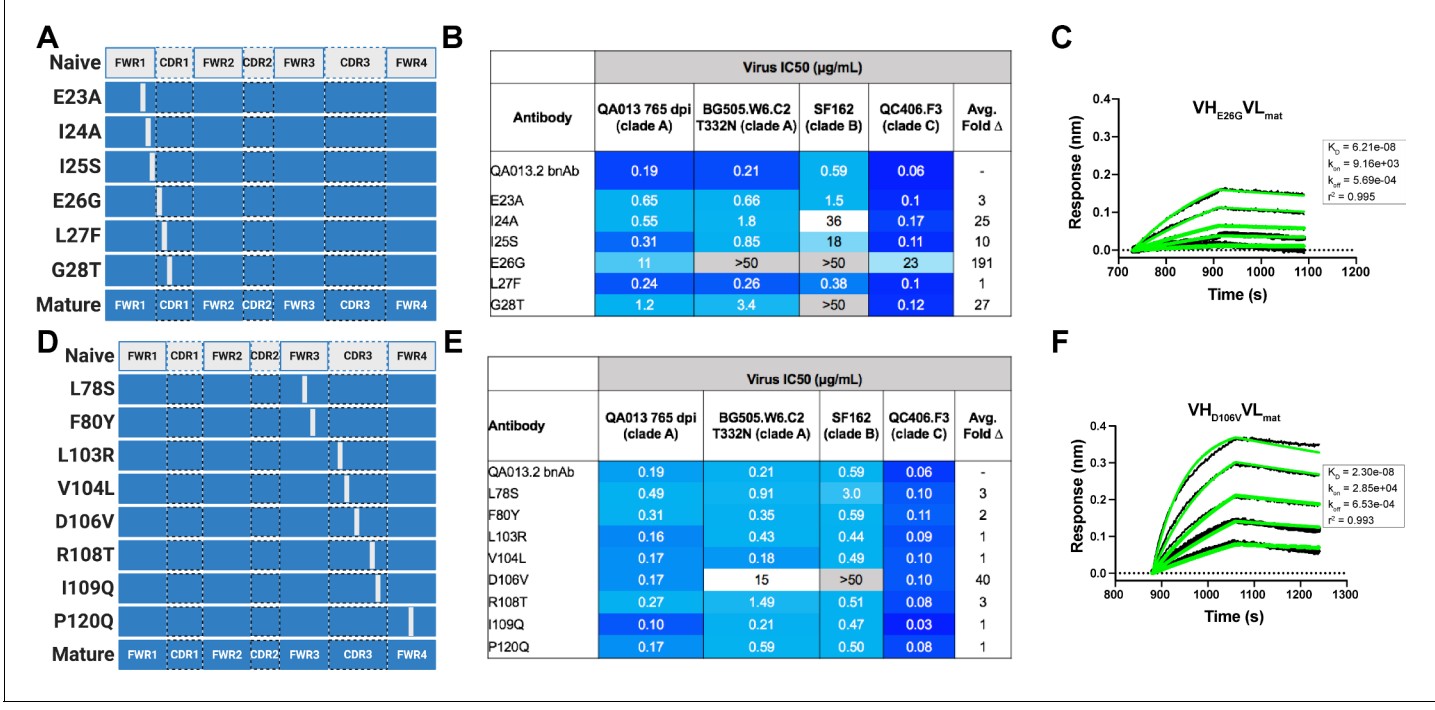

**Figure 5.** Binding and neutralization of VH point mutant antibodies demonstrates that residues spanning FWRH1-CDRH1 are essential for conferring bnAb neutralization breadth. Schematic of heavy chain point mutants generated in (**A**) FWRH1-CDRH1 that were paired with the mature light chain. Names of each heavy chain point mutant are listed on the left, with colored bars representing the VH region including FWR and CDR loops (dashed lines) colored by lineage member to visualize how each point mutant was generated. Inferred naive heavy chain is represented in gray and the mature heavy chain is shown in dark blue. (**B**) FWRH1-CDRH1 point mutant neutralization of a subset of autologous and heterologous pseudoviruses as measured by the TZM-bl assay. $IC_{50}$ values of each antibody are shown for each pseudovirus column, where darker blue indicates more potent neutralization and grey indicates no neutralization was observed at the highest antibody concentration tested. Average fold change in $IC_{50}$ for a particular point mutant relative to the mature bnAb across all pseudoviruses is reported as a final column on the right. (**C**) BLI kinetic curve of a selected FWRH1-CDRH1 point mutant antibody E26G at 10 µg mL$^{-1}$ against heterologous trimer (BG505.SOSIP.664 at 500 nM). (**D**) Schematic of CDRH3 mutants that were paired with the mature light chain. (**E**) CDRH3 point mutant neutralization of a subset of autologous and heterologous pseudoviruses as measured and displayed in panel B. (**F**) BLI kinetic curve of a selected CDRH3 point mutant antibody D106V at 10 µg mL$^{-1}$ against heterologous trimer (BG505.SOSIP.664 at 500 nM). BLI curves were analyzed as described in the *Figure 2* legend. See *Figure 5—source data 1*, *Figure 5—source data 2* and *Figure 5—source data 3*.See also *Supplementary file 3* for fasta file of all chimeric and mutant VH and VL antibody chains used in *Figures 4*, *5*, *6*.

The online version of this article includes the following source data for figure 5:

**Source data 1.** Biolayer Interferometry Source Data 1.
**Source data 2.** Biolayer Interferometry Source Data 2.
**Source data 3.** Neutralization Source Data.

## QA013.2 light chain confers higher bnAb potency

Although less impactful than the heavy chain, our aforementioned data suggested that light chain development indeed contributes to QA013.2 neutralization potency (*Figure 2*). To identify which regions of the light chain were essential for these contributions, we reverted regions of SHM within Int5$_{VL}$ back to naive (Int5ΔFWR2-FWR3$_{VL}$ and Int5ΔCDR3$_{VL}$) and paired these with the mature heavy chain (*Figure 6A*). Reversion of the residues spanning FWRL2-FWRL3 abrogated antibody binding to BG505.SOSIP.664 heterologous Env ($K_D$ = 1.3 µM) relative to the mature bnAb by more than 400-fold. This effect was less pronounced when residues in the CDRL3 were reverted to naïve (*Figure 6A,B*). These light chain variants also demonstrated reduced neutralization potency relative to the mature bnAb. Reversion of residues present in FWRL2-FWRL3 resulted in an average 19-fold increase in $IC_{50}$ (range 6–42 fold) and reversion of residues in CDRL3 caused a 14-fold increase in average $IC_{50}$ (range 1–41 fold) across all pseudoviruses tested when compared to the mature bnAb $IC_{50}$ values (*Figure 6A*). Together, these data demonstrate that residues in FWRL2-FWRL3 and

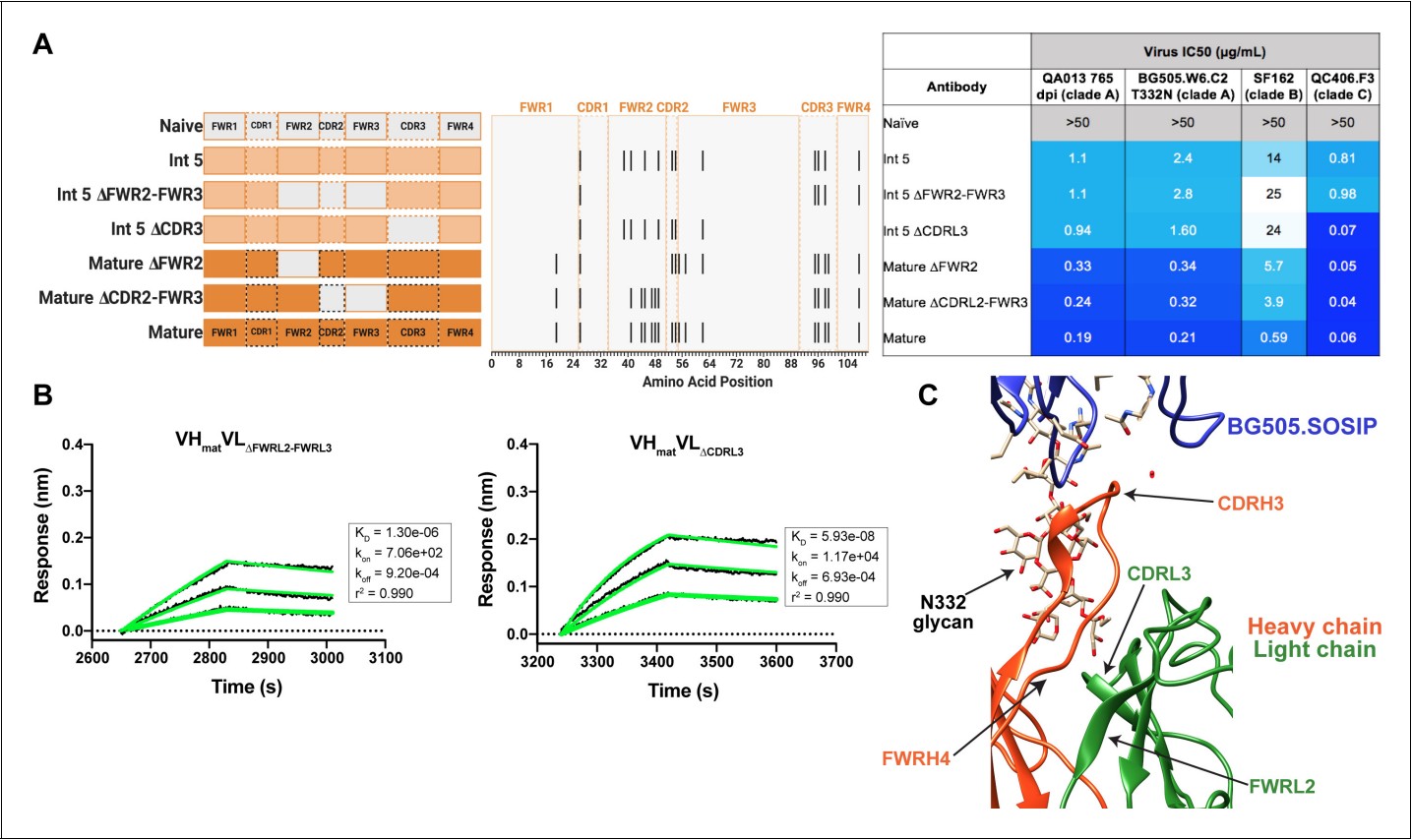

**Figure 6.** Construction of chimeric VL antibodies to investigate the contribution of QA013.2 light chain to bnAb breadth and potency. (**A**) Schematic of light chain intermediates that were generated and paired with the mature heavy chain. Names of each light chain intermediate are listed on the left, with colored bars representing the VL region including FWR and CDR loops (dashed lines) colored by lineage member to visualize how each variant was generated. Inferred naïve light chain is represented in gray, latest inferred light chain intermediate (Int5$_{VL}$) is shown in light orange, and the mature light chain is shown in dark orange. The plot in the middle shows the individual mutations across VL relative to the inferred naïve BCR, with FWR regions flanking CDR loops that are demarcated by dashed lines. Neutralization table on the far right shows the average IC$_{50}$ values for each tested intermediate against a panel of pseudoviruses grouped by clade shown at the top. Characteristics of the neutralization table are the same as those listed in the **Figure 2** legend. (**B**) Biolayer interferometry kinetic curves of a select subset of tested antibody variants (10 µg mL$^{-1}$ against heterologous trimer BG505.SOSIP.664 at 500 nM). BLI curves were analyzed as described in the **Figure 2** legend. All data are representative of at least two independent experiments. (**C**) Ribbon structure of QA013.2 Fab bound to BG505.SOSIP.664 Env trimer. Protruding N332 glycan on BG505.SOSIP.664 trimer is shown as a stick model, Fab heavy chain is shown in orange, and Fab light chain in green. Key regions of the structure are labeled including the N332 glycan as well as the VL regions that contribute to bnAb potency and antibody-antigen complex formation. FWRL2 interacts with FWRH4 of the heavy chain, while CDRL3 appears to interact with the tip of the N332 glycan. See **Figure 6—source data 1**, **Figure 6—source data 2** and **Figure 6—source data 3**. See also **Supplementary file 3** for fasta file of all chimeric and mutant VH and VL antibody chains used in **Figures 4**, **5**, **6**.

The online version of this article includes the following source data for figure 6:

**Source data 1.** Biolayer Interferometry Source Data 1.
**Source data 2.** Biolayer Interferometry Source Data 2.
**Source data 3.** Neutralization Source Data.

CDRL3, which are situated along the cleft between heavy and light chains and near the N332 glycan, contribute to the mature bnAb's binding affinity and neutralization potency, with residues in FWRL2-FWRL3 appearing to have a larger influence on bnAb binding. Notably, the residues spanning FWRL2-FWRL3 do not interact directly with the N332 glycan on Env. However, the residues in FWRL3 may influence the conformation of residues in the adjacent CDRL3 loop, which interacts with the N332 glycan. Furthermore, the residues in FWRL2 interact with FWRH4 of the Fab heavy chain (residues 117–120) which follows the CDRH3 loop (**Figure 6C**). Attempts to identify the light chain residues in FWRL2-FWRL3 responsible for mediating this interaction were inconclusive, as reversion of the residues located in FWRL2 and CDRL2-FWRL3, separately, did not influence the IC$_{50}$ values of

the tested pseudoviruses, indicating that the cumulative effect of these mutations was beneficial (*Figure 6A*).

## Mutational antigenic profiling of QA013.2 identifies unique viral escape mutations in Env variable loop 1

We used mutational antigenic profiling to interrogate the effect of all functionally tolerated single amino acid mutations in BG505.W6.C2 T332N Env on viral escape when under selective pressure from the mature QA013.2 bnAb (*Dingens et al., 2017*; *Figure 7*). In addition to the more traditional signatures of Env escape from V3/glycan-specific bnAbs, including mutations that disrupt the N301 and N332 potential N-linked glycosylation sites (PNGS) or alter the GDIR motif (*Dingens et al., 2019*), we also observed viral escape via mutations in the V1 loop (*Figure 7A,B*). In addition to the two PNGS, enrichment of viral escape mutations was strongest at sites D325, R327, and H330; validation of these data using pseudoviruses with individual mutations in TZM-bl neutralization assays showed that mutations to these sites can result in >32-fold change in IC$_{50}$ relative to the wild-type BG505.T332N (*Figure 7B,C*). However, not all mutations to these sites resulted in viral escape. Similar to 10–1074 (*Dingens et al., 2019*), while D325E was enriched in mutational antigenic profiling

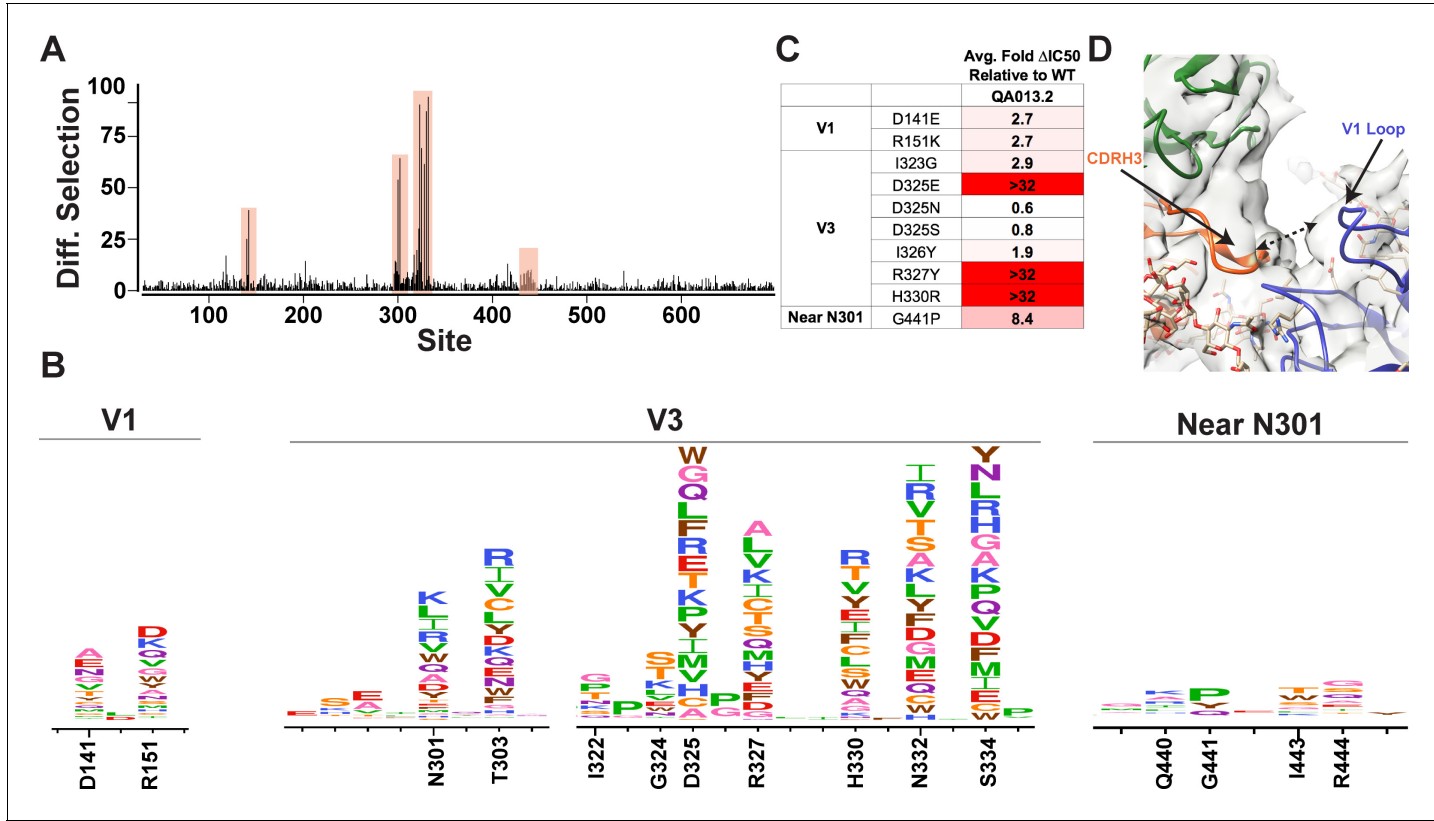

**Figure 7.** Mutational antigenic profiling of QA013.2 identifies unique viral escape mutations. (**A**) Line plot showing the median positive differential selection across the mutagenized portion of BG505.W6.C2 T332N Env. (**B**) Logoplots showing the mutation-level enrichment for areas of escape in A. (**C**) Table of average fold change in IC$_{50}$ for each BG505.W6.C2 T332N viral escape mutant relative to the wildtype BG505.W6.C2 T332N pseudovirus as tested in the TZM-bl neutralization assay. Darker red colors indicate a larger fold change in average IC$_{50}$. In vitro neutralization data are representative of three independent replicates. (**D**) Heavy chain CDR3 loop is within 8 Å distance to the V1 loop when measured between the Cα chains. See also *Figure 7—figure supplements 1* and *2* and *Figure 7—source data 1*.

The online version of this article includes the following source data and figure supplement(s) for figure 7:

**Source data 1.** Neutralization Source Data.

**Figure supplement 1.** Correlation of positive site differential selection between mutational antigenic profiling replicates.

**Figure supplement 2.** Mature bnAb QA013.2 binding to gp120 monomers that lack the N332 glycan.

**Figure supplement 2—source data 1.** Biolayer Interferometry Source Data 1.

**Figure supplement 2—source data 2.** Biolayer Interferometry Source Data 2.

and had a >32-fold effect in the TZM-bl assay, D325N/S mutants were not enriched and had negligible effects when tested individually via TZM-bl neutralization assay.

Viral escape in the V1 loop occurred at two sites that are adjacent in the BG505 structure; D141 and R151 (HXB2 numbering), both of which caused a modest ~3-fold change in $IC_{50}$ relative to the wildtype virus (*Figure 7B,C*). While the V1 loop of Env does not appear to be in direct contact with QA013.2 Fab in the complex structure, the backbone Cα atoms of the Fab CDRH3 loop and Env V1 loop are within 8 Å of each other (*Figure 7D*), suggesting there could be amino acid side-chain interactions between the two loops that can affect neutralization.

Interestingly, while the glycan at N332 is essential for bnAb neutralization and disrupting this motif results in viral escape, we observed that it is not required to achieve antibody binding – as demonstrated by the mature bnAb binding to the gp120s from the autologous clade D and BG505. W6.C1 viruses, both of which lack the PNGS at site 332 (*Figure 7—figure supplement 1*). It is likely that this binding, while measurable, is too weak to confer neutralization, confirming the importance of the N332 glycan.

## Influence of N301 glycan on QA013.2 bnAb neutralization

The QA013.2 bnAb has previously been shown to potently neutralize tier two viruses from clades A, B, and C, as well as several tier three viruses from clades B and C, but it did not neutralize clade D or A/D recombinant viruses (*Williams et al., 2018*). This study demonstrated the importance of the glycan at N332 for QA013.2-mediated neutralization, and this finding was corroborated in the cryo-EM structure, functional mutagenesis, and viral escape data that we present here (*Figures 3–5* and *7*). The structural data also suggest that the N301 glycan is playing a major role in facilitating antibody neutralization, based on its location adjacent to FWRH1-CDRH1 residues in the antibody VH region that we show are critical for bnAb breadth (*Figures 3–5*). Moreover, the antigenic profiling data highlight this glycan site as a key region facilitating viral escape in the V3 loop (*Figure 7*). However, the role of the glycan at N301 had not been examined directly in prior studies. To confirm the dependence of QA013.2 bnAb on the glycan at N301, we generated a N to K mutation at position 301 in BG505.W6.C2 T332N, which is the virus used for antigenic profiling, as K301 was identified as a major escape mutation (*Figure 7*). Elimination of the glycan at N301 abolished all neutralization activity despite the presence of the N332 glycan in BG505.W6.C2 T332N (*Figure 8*). These data demonstrate that both N301 and N332 glycans in combination are required to achieve bnAb neutralization.

We also tested the mature bnAb against an expanded panel of clade A and clade D pseudoviruses to better define the specificity of this bnAb with regard to the clade D initial infecting virus and clade A superinfecting virus. Similar to the findings with autologous virus, QA013.2 neutralized clade A but not clade D viruses; interestingly, the bnAb also did not neutralize D/A recombinant variants. Clade A virus neutralization was not universal, notably those viruses missing either glycan or the GDIR motif were not neutralized, which aligns with the predictions of key antigen contacts identified in the cryo-EM structure of QA013.2 bound to native-like Env trimer (*Figure 3*). The mature bnAb is only capable of neutralizing viruses that contain N301, N332, and the linear GDIR motif (*Figure 8*).

Interestingly, there are some viruses with all three of the V3 features that are not neutralized, which implies that there are other key residues involved. The antigenic profiling data demonstrate that charged residues in the V1 loop are also important for neutralization by QA013.2 (*Figure 7*). Indeed, we noted that all viruses that are neutralized by QA013.2 contain a (negative charge/ branched hydrophobic/positive charge) motif in the final 10 residues of the V1 loop, except MK184. W0.D1, which is a clade C/D recombinant with clade D V1-V5 sequence (*Wu et al., 2006*). The four viruses that contained the main antigenic contacts in Env (N301, N332, and GDIR) but were not neutralized by the mature bnAb were either subtype D in variable loops 1 and 2 (D/A recombinants; *Wu et al., 2006*), or they lacked this [negative charge/branched hydrophobic/positive charge] in V1. Overall, the neutralization results from this expanded panel demonstrates that the mature bnAb can only neutralize viruses from clades A, B, and C, not clade D (*Figure 8*), as suggested by studies using a smaller virus panel (*Williams et al., 2018*) and that V1 influences the antibody's function.

| Virus | Virus Characteristics | | | | | |
|---|---|---|---|---|---|---|
| | Clade | N301 glycan | N332 glycan | G(D/N)IR motif | V1/V2 Residues* | Avg. IC50 QA013.2 bnAb |
| BG505.W6.C2 T332N | A | ✓ | ✓ | ✓ | DDMRGELKNC | 0.21 |
| BG505.W6.C2 T332N N301K | A | | ✓ | ✓ | DDMRGELKNC | >50 |
| QA013.765M.C1 | A | ✓ | ✓ | ✓ | ANITTDMKNC | 0.19 |
| BG505.W6.C1 | A | ✓ | | ✓ | DDMRGELKNC | >50 |
| QB726.70M.B3 | A | ✓ | ✓ | ✓ | NEMPGEQNC | >50 |
| Q259.W6 | A | ✓ | | ✓ | NDMQEGKNC | >50 |
| QH343.21M.A10 | A | ✓ | ✓ | ✓ | PTVREDMKNC | 0.94 |
| QA255.21P.A15 | A | ✓ | ✓ | ✓ | TTIDKDMKNC | 3.1 |
| BI206.W6RT.A1 | A | ✓ | ✓ | ✓ | GKAREELRNC | 0.055 |
| BF520.W14.B3 | A | ✓ | ✓ | | NDMKEEITNC | >50 |
| BL274.W6.A3 | A | ✓ | ✓ | ✓ | MEMGQEIKNC | 0.33 |
| ML274.W0.D1 | A | ✓ | ✓ | ✓ | MEMGQEIKNC | 0.33 |
| MS208.W0.B.A3 | A | ✓ | ✓ | ✓ | VNMGEEVKNC | >50 |
| MI206.W0.A1 | A | ✓ | ✓ | ✓ | REAREELRNC | 0.55 |
| MJ613.W1.A2 | A | ✓ | ✓ | ✓ | DNEMKGEIKNC | 2.4 |
| MG505.W0.A2 | A | ✓ | | ✓ | DDMRGELKNC | >50 |
| MF535.W0.F1 | D/A | ✓ | | ✓ | NTTEAGMKNC | >50 |
| ML035.W0.A1 | D/A | ✓ | ✓ | ✓ | TGEDTGMKKC | >50 |
| BF535.W6.A1 | D/A | ✓ | | ✓ | NTTEAGMKNC | >50 |
| BL035.W6.B1 | D/A | ✓ | ✓ | ✓ | TGEDSGMKKC | >50 |
| SF162 | B | ✓ | ✓ | ✓ | EMDRGEIKNC | 0.59 |
| QC406.F3 | C | ✓ | ✓ | ✓ | MDMNGEIKNC | 0.060 |
| MK184.W0.D1 | C/D | ✓ | ✓ | ✓ | TISDIGMKNC | 0.11 |
| QA013.70I.H1 | D | ✓ | | ✓ | TDQDIGMKNC | >50 |
| QA465.59M.A1 | D | ✓ | ✓ | ✓ | TSEDTGMRNC | >50 |
| QB857.110I.B3 | D | ✓ | ✓ | ✓ | VTDDTGMRNC | >50 |

*Given the extreme variability in length of the V1/V2 loop, the reported residues include the final 10 amino acids at the end of V1 up to the conserved cysteine residue, before the start of V2.

**Figure 8.** QA013.2 bnAb neutralization of a virus panel including a virus lacking the N301 glycan. Tested pseudoviruses are shown with their respective clade in the left hand columns. Envelope characteristics are shown in the middle columns for each pseudovirus, including the presence (or absence) of glycans at N301 and N332 as well as the existence of the conserved G(D/N)IR linear motif at the base of the V3 loop as indicated by a check mark. The final ten residues in the V1 loop are shown for all pseudoviruses tested, in which residues that contain the [negative charge/branched hydrophobic/positive charge] motif are colored in red. The IC$_{50}$ for each tested pseudovirus is shown in a column on the far right of the table. Darker blue indicates more potent neutralization, while white demonstrates weak neutralization, and grey represents no neutralization observed at the highest antibody concentration tested. Simian immunodeficiency virus (SIV) was used as a negative control; the tested bnAb showed no evidence of SIV neutralization (data not shown). IC$_{50}$ values are the average of two independent replicates. See **Figure 8—source data 1**.
The online version of this article includes the following source data for figure 8:

**Source data 1.** Neutralization Source Data.

## Discussion

Vaccine approaches aimed at recapitulating the natural stimulation and maturation of bnAbs may require honing of the immune response through exposure to a series of diverse HIV immunogens. HIV-specific bnAbs have been shown to emerge more commonly in cases involving HIV superinfection (*Cortez et al., 2012*; *Powell et al., 2010*) and thus, provide a unique opportunity to identify the emergence of neutralization potency and breadth. Here, we investigated the functional development of V3/glycan-specific bnAb QA013.2, isolated from a case of HIV superinfection. QA013.2 uses unique gene segments (VH3-7*01 VD1-1*01 VJ5*02 and VL1-40*01 VJ2*01), has moderate levels of somatic hypermutation (≤ 20%), and unlike many HIV bnAbs, was not found to be polyreactive (*Figure 9—figure supplement 1*).

We present data detailing the lineage development and structure-function relationships of QA013.2 using antibody variable region deep sequencing and single particle cryo-EM respectively. From the deep sequencing data, we observed the presence of early bnAb lineage members circulating in the blood pre-HIV infection. This suggests that this antibody lineage may have been stimulated by an antigen other than HIV, as has been suggested for some gp41-specific antibodies, which recognize commensal bacteria in the gut (*Liao et al., 2011*; *Trama et al., 2014*; *Williams et al., 2015b*). Given the dependence of QA013.2 bnAb on the two glycans at N301 and N332 in combination, we speculate that this bnAb lineage may have been initiated by a glycosylated antigen prior to HIV infection. The high-resolution structure of QA013.2 Fab bound to the heterologous native-like Env trimer revealed specific regions of the bnAb paratope that interact with HIV Env, which guided targeted mutational approaches to identify the antibody determinants of potency and breadth. From our structural data and functional investigations, we demonstrate that while QA013.2 shares traits with other V3/glycan-specific bnAbs, such as targeting conserved glycans and the linear GDIR motif at the base of V3, it also relies less on the CDRH3 loop, which often plays a pivotal role in achieving bnAb neutralization breadth. Rather, residues spanning FWRH1-CDRH1 region were most critical for QA013.2 neutralization breadth. Thus, our results highlight universal aspects of the core V3 epitope on Env as well as the multitude of ways in which bnAbs evolve to target it.

Biolayer interferometry data collected on the QA013.2 bnAb lineage showed that in contrast to many other V3/glycan-specific bnAbs, the inferred naive BCR was capable of binding autologous gp120 monomer from the Env protein of the infecting HIV strain, in this case, the superinfecting strain. While we were unable to quantify the binding affinity of the inferred naive BCR, it has been estimated that affinities in the range of $\leq 1$ µM are needed to initiate naive BCR activation, with the frequency of the precursor B cells playing a critical role in the success of B cell activation, germinal center competition, and memory B cell production (*Abbott et al., 2018*). Despite precursor B cell frequencies in this subject not being known, the largest variable gene family, VH3, which QA013.2 bnAb uses (VH3-7), comprises nearly half of the VH repertoire in adult peripheral B cells (*Demaison et al., 1995*). Taken together, our data indicate that the bnAb lineage was likely stimulated by the second infection.

The neutralization profile of the mature bnAb supports the lineage being initiated by the superinfecting clade A virus, as we observed neutralization of both autologous and heterologous clade A viruses, but not clade D. In agreement with these data, the inferred naive BCR did not bind the autologous clade D Env from the initial infecting virus and thus did not appear to play a role in initiating the activation and expansion of the QA013.2 bnAb lineage. Interestingly, the mature bnAb was able to bind to the Env of this initial infecting virus, despite being unable to neutralize it. These data suggested that the development and affinity maturation of the QA013.2 bnAb lineage may have been influenced by both the initial and superinfecting viruses in this individual. Moreover, the presence of clade A/D recombinants in the viral population amplified as early as 385 dpi may have also played a role in this antibody's affinity maturation and development (*Williams et al., 2018*).

QA013.2 embodies several unique traits. Unlike other V3-targeting bnAbs such as PGT128 that have required significant modification of the paratope through insertions and deletions, we observed that antigen contacts here are mediated by a paratope that does not include indels but includes key framework residues in both the heavy (E23-I25) and light chains (FWRL2 and FWRL3) (*Figure 9*). In this case, we observed that without the acquired mutations in FWRL2-FWRL3, QA013.2 had a substantially slower $k_{on}$ rate, indicating that these residues contribute to bnAb affinity and antibody-antigen complex formation. The importance of framework substitutions to HIV bnAbs has been documented, in which they often provide flexibility that facilitates enhanced binding affinity and in some cases, interact directly with antigen (*Klein et al., 2013*). Additional involvement of framework regions in QA013.2 may result from a 'tilted' angle of approach adopted by the heavy chain of the Fab. A similar configuration was observed in 10–1074 and PGT121 bnAbs (*Mouquet et al., 2012*), In this case, the CDRH3 loop is tilted away from the rest of the heavy chain, toward the light chain resulting in a large portion of the CDRH3 loop being exposed at the antigen interface of the N332 glycan. We propose that this may facilitate additional interactions with this core glycan. The tilting also brings FWRH4 on the opposing side of the CDRH3 loop into close proximity with the light chain. Despite the minimal affinity maturation in this region, FWRH4 (residues 117–120) was found to interact with residues in the light chain FWRL2, likely stabilizing the conformations of the CDRH3 and CDRL3 loops, which in turn, contributes to the overall binding affinity of

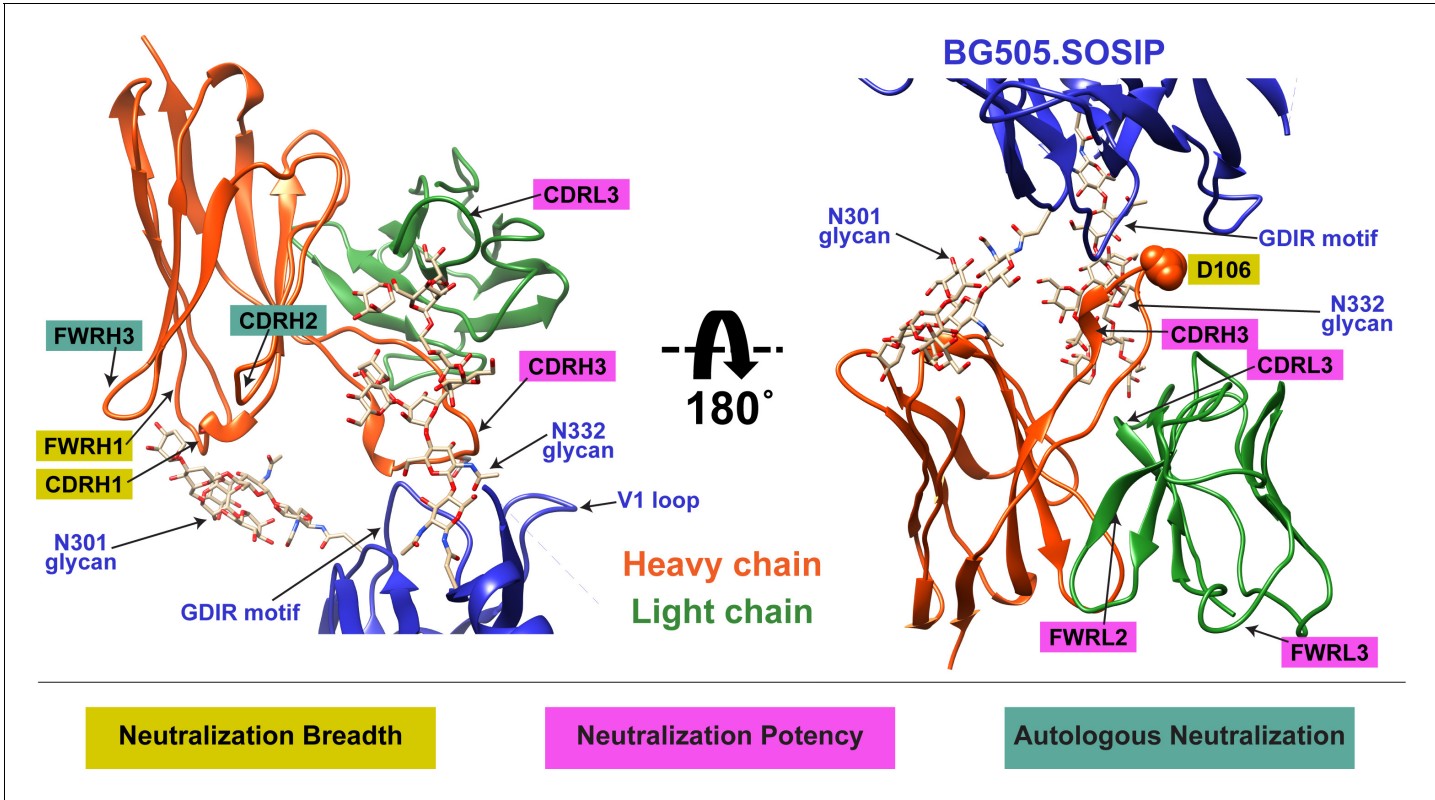

**Figure 9.** Summary of QA013.2 determinants of HIV neutralization breadth, potency, and autologous neutralization. Ribbon structure of QA013.2 Fab variable domain complexed to BG505.SOSIP.664 trimer in two different orientations are shown. Env structure is colored in blue with protruding glycans at sites N301 and N332 shown as stick models, Fab heavy chain in orange, and Fab light chain in green. Key regions of the Fab structure that confer breadth (yellow) or potency (pink) are labeled and colored categorically. Teal colored regions (CDRH2 and FWRH3) confer autologous neutralization. Important Env epitopes of interaction such as the linear GDIR motif and V1 loop are also labeled. See also *Figure 9—figure supplement 1*. The online version of this article includes the following figure supplement(s) for figure 9:

**Figure supplement 1.** QA013.2 bnAb polyreactivity as measured by ELISA.

this bnAb. In contrast to other V3-targeting bnAbs, QA013.2 required residues outside of the CDRH3 region in FWRH1-CDRH1 to achieve cross-clade neutralization breadth. Based on our targeted mutagenesis studies, the CDRH3 as a whole did not appear to mediate heterologous neutralization breadth. Only D106 alone was critical for cross-clade neutralization. The CDRH3 of the mature bnAb contains many hydrophobic residues, which could be implicated in Van der Waals interactions with the surface of Env or possibly in shielding water molecules from interacting with the charged residues of the GDIR motif, thus enabling D106 to interact with this linear motif instead. Alternatively, the charged D106 residue may be important for electrostatic interactions with the polar GDIR motif. From the cryo-EM structure, the FWRH1-CDRH1 region is adjacent to the conserved N301 glycan, highlighting this interaction as a possible driver of neutralization breadth. It's possible that the N332 glycan required for neutralization may have served as the initial anchor for the QA013.2 lineage and subsequent affinity maturation to accommodate and interact with N301 promoted neutralization breadth as has been previously reported for sialic-acid-bearing glycans and VRC01-class bnAbs (*Andrabi et al., 2017*; *IAVI Protocol C Investigators et al., 2019*, p. 01). These findings may inform future immunogen design aimed at eliciting V3/glycan-specific bnAbs as they illuminate structural regions of the bnAb paratope that may promote neutralization breadth and increased potency when modified appropriately.

The ability of HIV to escape recognition by neutralizing antibodies in vivo is well documented (*Bonsignori et al., 2017b*, pp. 10–10; *Caskey et al., 2017*, pp. 10–1074; *Klein et al., 2012*; *Scheid et al., 2016*; *Wei et al., 2003*; *Wibmer et al., 2013*) and can be comprehensively measured using antigenic profiling with a library of all possible amino acid mutations across HIV Env

(*Dingens et al., 2019*; *Dingens et al., 2017*). Using this technique, we found that QA013.2 elicited viral escape mutations throughout the V3 loop of Env. Unlike V3-bnAbs BF520.1 and 10–1074 that have more focused escape signatures concentrated on a handful of sites within V3, QA013.2 elicited escape mutations from more sites across this variable region of Env (*Dingens et al., 2019*). Selection for escape mutations was strongest at sites D325, N332, and S334, which supports the documented viral escape that occurred in this patient in vivo, resulting in an observed glycan shift from site N332 to N334 (*Williams et al., 2018*). Moreover, these data highlight the functional importance of the GDIR motif residues in mediating bnAb binding via the CDRH3. QA013.2 also selected for escape mutations outside of V3, in nearby V1 at sites 141 and 151, which are separated by a single residue in BG505 Env due to a shorter V1 loop relative to the reference strain. While we do not observe direct interaction between the Fab and Env V1 loop based on our cryo-EM structure, the main chain of the CDRH3 loop is in close proximity to the V1 loop, suggesting possible side chain electrostatic interactions between residues comprising the two regions. In vivo data have demonstrated evidence of V1-mediated escape from members of the DH270 antibody lineage and structural studies of the PGT121 family of bnAbs and BG18 have demonstrated that these antibodies interact with residues and glycans in this region (*Barnes et al., 2018*; *Bonsignori et al., 2017a*; *Garces et al., 2015*). Moreover, there is evidence supporting an inverse correlation between V1 loop length and bnAb breadth (*Barnes et al., 2018*; *Bonsignori et al., 2017a*; *Fera et al., 2018*; *Liu et al., 2011*; *Rusert et al., 2011*; *Saunders et al., 2019*). There are also data supporting a relationship between V1/V2 and V3, in which mutations in V1/V2 were shown to influence the 'flickering' of V3 between occluded and available conformational states (*Powell et al., 2017*; *Zolla-Pazner et al., 2016*). Together, these data suggest that immunogens with a relatively short ($\leq$ 28 residues) V1 loop comprised of polar residues at sites 141 and 151 may stimulate a similar neutralization profile to that of QA013.2. It is also possible that immunogens with even shorter V1 loops than studied here ($\leq$ 18 residues) may inhibit the electrostatic interactions we hypothesize for QA013.2 and prevent the possibility of viral escape in this region, likely leading to increased breadth.

Collectively, our study reveals several features of V3/glycan-specific bnAb development in a case of HIV superinfection. These include paratope differences for autologous and heterologous neutralization as well as non-overlapping determinants of neutralization breadth and potency providing a potential mechanism by which distinct genetically diverse immunogens contribute to such results. In addition, our data highlight features of the conformation-dependent V3 epitope that could be valuable for immunogen design, including short V1 regions and accommodation of the conserved glycan at N301 as a potential driver of neutralization breadth. Designing immunogens that lack nearby obstructive Env features may influence nAb accommodation of N301 and drive subsequent breadth as has been shown previously for a VRC01-class bnAb and the N276 glycan (*IAVI Protocol C Investigators et al., 2019*, p. 01). In summary, our findings suggest a strategy that promotes immunogen recognition via conserved V-gene features and stimulation of framework substitutions in order to elicit potent and broad neutralizing antibodies targeting the V3 supersite of vulnerability.

## Materials and methods

### Key resources table

| Reagent type (species) or resource | Designation | Source or reference | Identifiers | Additional information |
|---|---|---|---|---|
| Strain, strain background (HIV-1) | HIV Env clone BG505. W6M.C2 T332N | *Wu et al., 2006* | GenBank: DQ208458.1 | |
| Strain, strain background (HIV-1) | HIV Env clone SF162 | *Cheng-Mayer et al., 1997* | GenBank: EU123924.1 | |
| Strain, strain background (HIV-1) | HIV Env clone QC406.70M.F3 | *Blish et al., 2009* | Genbank: FJ866133.1 | |

*Continued on next page*

*Continued*

| Reagent type (species) or resource | Designation | Source or reference | Identifiers | Additional information |
|---|---|---|---|---|
| Strain, strain background (HIV-1) | HIV Env clone QA013.765M.C1 | *Williams et al., 2018* | Genbank: MG992331 | |
| Strain, strain background (HIV-1) | QB726.70M.B3 | *Blish et al., 2009* | Genbank: FJ866111.1 | |
| Strain, strain background (HIV-1) | Q259.W6 | *Long et al., 2002* | Genbank: AF407151.1 | |
| Strain, strain background (HIV-1) | QH343.21M.A10 | *Blish et al., 2009* | Genbank: FJ866119.1 | |
| Strain, strain background (HIV-1) | QA255.21P.A15 | *Bosch et al., 2010* | Genbank: MW383929.1 | |
| Strain, strain background (HIV-1) | BI206.W6RT.A1 | *Wu et al., 2006* | Genbank: DQ208465.1 | |
| Strain, strain background (HIV-1) | BF520.W14.B3 | *Simonich et al., 2016* | Genbank: KX168093.1 | |
| Strain, strain background (HIV-1) | BL274.W6.A3 | *Wu et al., 2006* | Genbank: DQ208499.1 | |
| Strain, strain background (HIV-1) | ML274.W0.D1 | *Wu et al., 2006* | Genbank: DQ208495.1 | |
| Strain, strain background (HIV-1) | MS208.W0.B.A3 | *Wu et al., 2006* | Genbank: DQ187022.1 | |
| Strain, strain background (HIV-1) | MI206.W0.A1 | *Wu et al., 2006* | Genbank: DQ208459.1 | |
| Strain, strain background (HIV-1) | MJ613.W1.A2 | *Wu et al., 2006* | Genbank: DQ208444.1 | |
| Strain, strain background (HIV-1) | MG505.W0.A2 | *Wu et al., 2006* | Genbank: DQ208449.1 | |
| Strain, strain background (HIV-1) | MF535.W0.F1 | *Wu et al., 2006* | Genbank: DQ208429.1 | |
| Strain, strain background (HIV-1) | ML035.W0.A1 | *Wu et al., 2006* | Genbank: DQ208468.1 | |
| Strain, strain background (HIV-1) | BF535.W6.A1 | *Wu et al., 2006* | Genbank: DQ208431.1 | |
| Strain, strain background (HIV-1) | BL035.W6.B1 | *Wu et al., 2006* | Genbank: DQ208479.1 | |
| Strain, strain background (HIV-1) | MK184.W0.D1 | *Wu et al., 2006* | Genbank: DQ208484.1 | |

*Continued on next page*

*Continued*

| Reagent type (species) or resource | Designation | Source or reference | Identifiers | Additional information |
|---|---|---|---|---|
| Strain, strain background (HIV-1) | QA465.59M.A1 | *Blish et al., 2009* | Genbank: FJ866136.1 | |
| Strain, strain background (HIV-1) | QB857.110I.B3 | *Blish et al., 2009* | Genbank: FJ866138.1 | |
| Strain, strain background (*Escherichia coli*) | One Shot Top10 | Thermo Fisher | Cat#C404003 | chemically competent cells |
| Cell line (*Homo sapiens*) | FreeStyle 293F | Invitrogen | Cat#R790-07; RRID:CVCL_D603 | |
| Cell line (*Homo sapiens*) | Human Embryonic Kidney (HEK) 293T | ATCC | Cat# CRL-3216; RRID:CVCL_0063 | |
| Cell line (*Homo sapiens*) | TZM-bl | AIDS Reagent Program, Division of AIDS, NIAID, NIH. | Cat# ARP-8129; RRID:CVCL_B478 | |
| Biological sample (*Homo sapiens*) | Human PBMC samples from subject QA013 | *Martin et al., 1998* | N/A | |
| Antibody | VRC01 (human, monoclonal) | *Williams et al., 2019* | N/A | ELISA, tested at 25 µg/mL |
| Antibody | NIH 45–46 (human, monoclonal) | *Scheid et al., 2009* | N/A | ELISA, tested at 25 µg/mL |
| Antibody | 8ANC195 (human, monoclonal) | *Scheid et al., 2011* | N/A | ELISA, tested at 25 µg/mL |
| Antibody | 4E10 (human, monoclonal) | *Stiegler et al., 2001* | N/A | ELISA, tested at 25 µg/mL |
| Antibody | Fi6v3 (human, monoclonal) | Jesse Bloom | N/A | ELISA, tested at 25 µg/mL |
| Recombinant DNA reagent | Human Igγ1 expression vector | *Tiller et al., 2008* | Addgene: 80795 | |
| Recombinant DNA reagent | Human Igκ expression vector | *Tiller et al., 2008* | Addgene: 80796 | |
| Recombinant DNA reagent | Human Igλ expression vector | *Tiller et al., 2008* | Addgene: 99575 | |
| Peptide, recombinant protein | BG505.SOSIP.664 | *Sanders et al., 2013*; *Verkerke et al., 2016* | Courtesy of Kelly Lee | |
| Peptide, recombinant protein | Protein G agarose | Pierce | Cat#20397 | |
| Peptide, recombinant protein | BG505.W6.C1 gp120 monomer | Immune Technology Corp. | Cat#IT-001–176 p | |
| Peptide, recombinant protein | QA013.70I.H1 gp120 monomer | Cambridge Biologics | GenBank: MG992347 | custom protein synthesis |
| Peptide, recombinant protein | QA013.765M.C1 gp120 monomer | Cambridge Biologics | Genbank: MG992331 | custom protein synthesis |
| Commercial assay or kit | Gal-Screen | Thermo Fisher | Cat#T1028 | |
| Commercial assay or kit | AllPrep DNA/RNA Mini Kit | Qiagen | Cat#80204 | |
| Commercial assay or kit | SMARTer RACE 5'/3' Kit | Takara Bio USA | Cat#634858 | |
| Commercial assay or kit | KAPA library quantification kit | Kapa Biosystems | Cat#KK4824 | |

*Continued on next page*

*Continued*

| Reagent type (species) or resource | Designation | Source or reference | Identifiers | Additional information |
|---|---|---|---|---|
| Commercial assay or kit | 600-cycle MiSeq Reagent Kit v3 | Illumina | Cat#MS-102–3003 | |
| Commercial assay or kit | Nextera XT 96-well index kit | Illumina | Cat#FC-131–1001 | |
| Commercial assay or kit | QIAquick PCR Purification Kit | Qiagen | Cat#28104 | |
| Chemical compound, drug | FreeStyle MAX | Thermo Fisher | Cat#16447500 | |
| Chemical compound, drug | 293F FreeStyle Expression media | Invitrogen | Cat#12338–026 | |
| Chemical compound, drug | Q5 High-Fidelity Master Mix | New England BioLabs | Cat#M0492S | |
| Chemical compound, drug | DEAE-dextran | VWR | Cat#97061–684 | |
| Software, algorithm | FLASH v1.2.11 | *Magoč and Salzberg, 2011* | http://ccb.jhu.edu/software/FLASH/ | |
| Software, algorithm | Cutadapt 1.14 with Python 2.7.9 | *Martin, 2011* | http://cutadapt.readthedocs.io/en/stable/ | |
| Software, algorithm | FASTX toolkit 0.0.14 | Hannon Lab, Cold Spring Harbor | http://hannonlab.cshl.edu/fastx_toolkit/ | |
| Software, algorithm | Partis | *Ralph and Matsen, 2016* | https://github.com/psathyrella/partis | |
| Software, algorithm | FastTree 2 | *Price et al., 2010* | https://github.com/matsengrp/cft/blob/master/bin/prune.py | |
| Software, algorithm | Linearham | *Dhar et al., 2020* | https://github.com/matsengrp/linearham | |
| Software, algorithm | RevBayes | *Höhna et al., 2016, Höhna et al., 2021* | https://revbayes.github.io/ | |
| Software, algorithm | Local BLAST | Biopython | https://github.com/matsengrp/cft/blob/master/bin/blast.py | |
| Software, algorithm | WebLogo | *Crooks et al., 2004* | RRID:SCR_010236 | |
| Software, algorithm | Geneious v11.1.2 | *Kearse et al., 2012* | RRID:SCR_010519 | |
| Software, algorithm | Excel | Microsoft | RRID:SCR_016137 | |
| Software, algorithm | ForteBio's Octet Software 'Data Analysis 7.0' | Pall ForteBio (now Sartorius) | N/A | |
| Software, algorithm | Prism 9.0 | GraphPad | RRID:SCR_002798 | |
| Software, algorithm | Relion v3.0.7 | *Scheres, 2012* | RRID:SCR_016274 | |
| Software, algorithm | MotionCor2 | *Zheng et al., 2017* | RRID:SCR_016499 | |
| Software, algorithm | CTFFIND4 | *Rohou and Grigorieff, 2015* | RRID:SCR_016732 | |
| Software, algorithm | UCSF Chimera | *Pettersen et al., 2004* | RRID:SCR_004097 | |
| Software, algorithm | SWISS-MODEL | *Waterhouse et al., 2018* | RRID:SCR_013032 | |

*Continued on next page*

*Continued*

| Reagent type (species) or resource | Designation | Source or reference | Identifiers | Additional information |
|---|---|---|---|---|
| Software, algorithm | Phenix software suite | *Liebschner et al., 2019* | RRID:SCR_014224 | Autosharpen program (*Terwilliger et al., 2018*); 'Dock In Map' program; Realspace RefinementOo |
| Other | Anti-human IgG Fc capture biosensors | Sartorius | Cat#18–5063 | |
| Other | QA013 antibody variable gene sequencing data | This paper | BioProject SRA: PRJNA674442 | |
| Other | cryo-EM map and atomic coordinates | This paper | EMD-24195 and PDB 7N65 | |

## Human subjects

Peripheral blood mononuclear cell (PBMC) samples were collected between 1994–1997 from female subject QA013 at various times pre- and post-HIV infection. QA013 was enrolled in a prospective cohort of HIV negative, high-risk women in Mombasa, Kenya and was 30 years old at the time of her enrollment in 1993 (*Martin et al., 1998*). During the study, QA013 seroconverted following initial infection with a clade D virus (1995), and was subsequently superinfected with a clade A virus between 265 and 385 days post initial infection (*Chohan et al., 2005*). Study participants were treated according to Kenyan National Guidelines and did not receive any antiretroviral treatment during the time period in which the samples used in this study were collected. Antiretroviral therapy was offered as of March 2004, with support from the President's Emergency Plan for AIDS Relief. This study was approved by members of the ethical review committees at the University of Nairobi, the Fred Hutchinson Cancer Research Center, and the University of Washington. Study participants provided written informed consent prior to enrollment.

## Cell lines

For antibody production: HEK 293 F cells (RRID:CVCL_6642; originally derived from female human embryonic kidney cells) were obtained from Invitrogen (Thermo Fisher Scientific, Waltham, MA, catalog #R790-07) and grown at 37°C in Freestyle 293 Expression Medium (Thermo Fisher Scientific, catalog #12338002) in baffle-bottomed flasks rotating at 135 rpm. These cells were tested for mycoplasma and found to have no contamination. We verified the identity of these cells through short tandem repeat (STR) DNA profiling and found that they are 92% identical to HEK 293-FT cells (RRID:CVCL_6911), which are derived from the parent HEK 293-F cells (RRID:CVCL_6642) based on 15 markers.

For pseudovirus production: 293 T cells (RRID:CVCL_1926; a transformed cell line originally derived from female human embryonic kidney cells) were obtained from ATCC (Manassas, VA, catalog #CRL-3216) and grown at 37°C in DMEM media with added fetal bovine serum (10%), penicillin (10,000 units/mL), streptomycin 10,000 µg/mL, and Amphotericin B (250 ng/mL) (all from Thermo Fisher Scientific). These cells were tested for mycoplasma and found to have no contamination. We verified the identity of these cells through short tandem repeat (STR) DNA profiling and found that they are 91% identical to Anjou 65 cells (RRID:CVCL_3645), which are derived from the parent HEK 293T/17 cells (RRID:CVCL_1926) based on 8 markers.

For neutralization assays: TZM-bl cells (RRID:CVCL_B478; originally derived from female cancerous human cervical tissue) were obtained from NIH AIDS Reagent Program (Germantown, MD, catalog #ARP-8129) and grown at 37°C in DMEM media with added fetal bovine serum (10%), penicillin (10,000 units/mL), streptomycin 10,000 µg/mL, and Amphotericin B (250 ng/mL) (all from Thermo Fisher Scientific). These cells were tested for mycoplasma and found to have no contamination. We verified the identity of these cells through short tandem repeat (STR) DNA profiling and found that they are 93% identical to HeLa cells (RRID:CVCL_0030) based on 15 markers.

## RNA isolation, library preparation, and full-length antibody gene deep sequencing

Four longitudinal PBMC samples from 379 days pre-HIV infection (D-379), D-351, 385 days post initial infection (D385), and D765 were thawed in a 37°C water bath, diluted 10-fold in RPMI, and pelleted at 300 x *g* for 10 min. Cell pellets were washed with 1X phosphate-buffered saline (PBS), counted using trypan blue solution, centrifuged, and then total RNA was extracted using the AllPrep DNA/RNA Mini Kit (Qiagen, Germantown, MD) according to the manufacturer's protocol. Isolated RNA was stored at −80 °C.

Antibody sequencing was performed as previously described (*Simonich et al., 2019*; *Vigdorovich et al., 2016*) using the SMARTer RACE 5'/3' Kit (Takara Bio USA Inc, Mountainview, CA). Briefly, RACE-ready cDNA synthesis was performed using primers specific for IgM, IgG, IgK, and IgL as previously described (*Simonich et al., 2019*). We used molecular barcoding to eliminate errors introduced during library preparation and sequencing, which facilitated accurate measurement of genetic diversity and improved ancestral sequence reconstruction. Template switch adaptor primers that included unique molecular identifiers were used in the cDNA synthesis step: SmartNNNa 5' AAGCAGUGGUAUCAACGCAGAGUNNNNNUNNNNNUNNNNNUCTTrGrGrGrGrG 3' (*Turchaninova et al., 2016*), where 'rG' indicates ribonucleoside guanosine bases. Following synthesis, cDNA was diluted in 10 mM Tricine-EDTA according to the manufacturer's recommended protocol. First-round PCR of 14 cycles was performed using Q5 High-Fidelity Master Mix (New England BioLabs, Ipswich, MA) and nested gene-specific primers at 20 µM. First-round amplicons were used as templates for second-round PCR amplification (six cycles), which added MiSeq adaptors. Second-round products were gel purified and indexed using Nextera XT P5 and P7 indices (Illumina, San Diego, CA). Gel-purified indexed libraries were quantified using the KAPA library quantification kit (Kapa Biosystems, Wilmington, MA) using an Applied Biosystems 7500 real-time PCR machine. Libraries were denatured and loaded onto an Illumina MiSeq using 600-cycle V3 cartridges, according to the manufacturer's instructions.

## Sequence analysis and clonal family clustering

Following next-generation sequencing, sequence reads were pre-processed into amplicons using FLASH, primers were trimmed using cutadapt, and the FASTX-toolkit was used to remove sequence reads containing low-confidence base calls (N's) as previously described (*Simonich et al., 2019*; *Vigdorovich et al., 2016*). Sequence reads from each timepoint were merged together into an aggregate dataset that was deduplicated and annotated using *partis* (https://github.com/psathyrella/partis) with default settings. Sequences that were out of frame or contained internal stop codons were removed, while singletons, or sequences that were observed only once in the sampled repertoire, were included in an attempt to retain undersampled or rare sequences. Sequencing statistics for this study can be found in *Table 1*.

Clonal family clustering was performed on both individual timepoint and aggregated datasets using both partis seeded and unseeded clustering methods (*Ralph and Matsen, 2016*). The seeded method extracts all sequences that are clonally related to a particular 'seed' sequence of interest. The 'seeds' here refer to the previously-isolated QA013.2 VH and VL sequences (*Williams et al., 2018*). Unseeded clustering, on the other hand, clusters all sequences in the repertoire into clonal families. Since this is more computationally demanding, some of the very large datasets were subsampled down to 50–150 k sequences. Three random subsamples were analyzed and compared in order to measure statistical uncertainties. No subsampling was done for the seeded analyses.

Because the light chain has much lower diversity than the heavy chain, and because B cells proliferate after successful heavy chain rearrangement but before light chain rearrangement, light chain clustering without heavy/light chain pairing information inevitably incorrectly clusters together many different light chain families. This overclustering artificially inflates the number of sequences in the clonal family, resulting in most inferred light chain families containing sequences stemming from several different light chain rearrangements.

## Naive inference and lineage reconstruction

The light chain family resulting from clonal family clustering was very large (1158 members), and because this apparent family was likely composed of sequences from several different light chain

rearrangements, each of which are likely confined to separate lineages, a pruning procedure was performed based on phylogenetic relatedness to the QA013.2 antibody chain using FastTree 2 (*Price et al., 2010*). This resulted in the light chain clonal family being reduced to the most relevant 100 sequences (https://github.com/matsengrp/cft/blob/master/bin/prune.py). The VH clonal family did not undergo any such pruning as it contained fewer than 100 sequences (note that the tenfold different in size between VH and VL clonal families is itself strong evidence for overclustering).

The heavy and pruned light chain clonal families were then analyzed with *linearham* (https://github.com/matsengrp/linearham), a phylogenetic hidden Markov model that accounts for both VDJ recombination and SHM when inferring ancestral sequences including the inferred naive B cell receptor (BCR) (*Dhar et al., 2020*). Partis software also infers the unmutated common ancestor, or naive sequence; however, the version of partis used here assumed a balanced, star-like clonal family tree architecture, and the alternative naive sequences it returns are not based on a rigorous probabilistic model (current versions of partis no longer assume star-like phylogenies). Linearham is thus more accurate for trees, and also returns proper uncertainty estimates, because it samples naïve sequences from their posterior distribution and accounts for uncertainty in the naive rearrangement process.

As part of the linearham package, RevBayes (*Höhna et al., 2016*) was run using an unrooted tree model with the general time-reversible (GTR) substitution model. Each RevBayes run had customized settings to ensure that likelihood and posterior estimated sample sizes were >100 for the heavy and light chain lineages. For both VH and VL lineage inference, MCMC iterations were 100,000 with a thinning frequency of 100 iterations and 10,000 burn-in samples. Linearham generates summary graphics following phylogenetic inference that display only the edges that satisfy the given posterior probability threshold, and only the nodes that contact edges above the threshold. These graphics detail the relative confidence in unique inferred sequences and amino acid substitutions (*Figure 1* and *Figure 1—figure supplements 2* and *3*). Inferred intermediate sequences present in the most probable lineage paths were selected for study (see *Figure 1*).

## Identification of lineage-representative NGS sequences

To determine whether the computationally inferred naive BCR and intermediate ancestor sequences for VH and VL lineages were observed in the sampled NGS sequences, we ran the following script: https://github.com/matsengrp/cft/blob/master/bin/blast.py. A local BLAST database was created for the two seeded clonal families and queried for sampled NGS sequences that had high nucleotide sequence identity to each of the VH and VL lineage members using the 'blastn' command (Biopython package). E-value of 0.001 was used; other settings were default. Blastn 'hits' for each lineage member were sorted according to their percent nucleotide identity and alignment length. Percent nucleotide identity of the top blastn 'hit' for each lineage member is listed in *Figure 1*.

## Monoclonal antibody generation and mutagenesis

Antibody heavy and light chain sequences were synthesized as 'fragmentGENES' by Genewiz (GEN-EWIZ, South Plainfield, NJ) or gene blocks by Integrated DNA Technologies (IDT, Coralville, IA) and cloned into IgG1 and IgL vectors as previously described (*Williams et al., 2015a*). Human embryonic kidney (HEK) 293 F cells (RRID:CVCL_6642; Thermo Fisher Scientific, Waltham, MA, catalog #R790-07) were transfected with equal amounts of heavy and light chain plasmid DNA using the Freestyle Max transfection system (Thermo Fischer Scientific, catalog #16447100) according to the manufacturer's instructions. Transfections proceeded at 37°C with 8% $CO_2$ in baffled-bottom flasks, rotating at 135 rpm and were harvested after 3–6 days. Supernatants were collected following centrifugation of cells at 1900 rpm for 10 min. Antibody supernatants were passed three times over columns packed with immobilized Protein G resin (Thermo Fisher, catalog #20397) as previously described (*Scherer et al., 2014*).

Antibody chimeras were synthesized as fragmentGENES as shown in *Figure 4A*, while VH mutagenesis (*Figure 5*) was performed using phosphorylated primers (25 µM) that introduced the mutation of interest into the variable heavy chain, using mature QA013.2 VH DNA in the IgG1 plasmid as a template (2.5 ng/µL). PCR-mediated mutagenesis was performed using Phusion Hi Fidelity polymerase (Thermo Fisher Scientific, catalog #F-530) with the following thermocycler settings: denature at 98°C for 1 min, 30 cycles of 98°C for 30 s, annealing at 55–70°C for 30 s, extension at 72°C for 5

min, and final extension at 72°C for 10 min. PCR products were digested with DpnI (New England BioLabs, catalog # R0176) overnight at 37°C and purified using Qiagen PCR Purification kit (Qiagen, catalog #28104), with products eluted in ddH$_2$0. Mutants were ligated at room temperature for at least two hours prior to transformation into One Shot TOP10 chemically competent cells (Thermo Fisher Scientific, catalog #C404003). Mutant antibody chains were sequence-confirmed prior to transfection and antibody purification as described above.

## Pseudovirus production and TZM-bl neutralization assay

HIV pseudovirus (PSV) was produced in HEK 293 T cells (ATCC, Manassas, VA, CRL-3216) and used in TZM-bl neutralization assays as previously described (*Goo et al., 2012*). Briefly, monoclonal antibodies were incubated with PSV at 37°C for 1 hr and TZM-bl cells (obtained through the NIH AIDS Reagent Program, Germantown, MD, catalog #8129 from Dr. John C. Kappes, and Dr. Xiaoyun Wu) were added at 10,000 cells/well and the assay was harvested after 48 hr at 37°C by adding Gal-Screen substrate (Thermo Fisher Scientific, catalog #T1028) as previously described (*Simonich et al., 2019*). IC$_{50}$ values represent the half maximal inhibitory concentration in ug mL$^{-1}$ at which 50% of the virus was neutralized. All neutralization data are the average of at least two independent experiments, each performed in technical duplicate (see *Figure 2—source data 1*; *Figure 4—source data 1*; *Figure 5—source data 1*; *Figure 6—source data 1*; *Figure 7—source data 1*; *Figure 8—source data 1*). IC$_{50}$ values that deviated from the technical replicates by >10-fold were excluded as outliers from the average IC$_{50}$ calculation.

## Biolayer interferometry

QA013 monoclonal antibody binding was measured using biolayer interferometry on an Octet RED instrument (Forte Bio, Fremont, CA). All antibodies were diluted to 10 μg mL$^{-1}$ in a 0.22 μm filtered binding buffer solution composed of 1X PBS, 0.1% bovine serum albumin, 0.005% Tween-20, and 0.02% sodium azide. Antibodies were immobilized onto anti-human IgG Fc capture biosensors (Sartorius, Göttingen, Germany, catalog #18–5063) and dipped into analyte solution of several different antigens as detailed in the text (*Figures 2* and *4–6*): autologous gp120 monomers from the initial infecting virus Env (clade D) at 2 μM, the superinfecting virus variant from 765 days post initial infection (clade A) at 2 μM, heterologous BG505.SOSIP.664 Env trimer containing the N332 glycan (clade A) at 500 nM, or wildtype BG505.W6.C1 gp120 monomer at 2 μM. Antigens were diluted to their respective starting concentrations in the same buffer as above and a series of six, twofold dilutions were tested as analyte in solution at a shake speed of 600 rpm at 23°C. The kinetics of mAb binding were measured as follows: association was monitored for 180 s, dissociation was monitored for 180 s, and regeneration was performed in 10 mM Glycine HCl (pH 1.5, GE Healthcare, Chicago, IL, BR-1003–54). Binding-affinity constants (K$_D$; on-rate, k$_{on}$; off-rate, k$_{dis}$) were calculated using ForteBio's Data Analysis Software 7.0. Responses (nanometer shift) were calculated using data that were double reference subtracted using reference wells and non-specific binding of biosensor to analyte. Data were processed by Savitzky-Golay filtering, prior to fitting using a 1:1 model of binding kinetics. Prism v9.0 (GraphPad, San Diego, CA) was used to plot both the processed data (following double reference subtraction) and fit data (after global analysis) of each antibody + antigen combination. Each biolayer interferometry experiment presented in this study was performed a minimum of two independent times, each time using freshly diluted antibodies in binding buffer (see *Figures 2* and *4–6* – BLI Source Data 1 and 2). Both QA013 gp120 monomers were synthesized as custom proteins through Cambridge Biologics (Brookline, MA). BG505.SOSIP.664 trimer was expressed and purified as previously described (*Verkerke et al., 2016*) and the BG505.W6.C1 gp120 protein was ordered through Immune-Tech (Immune Technology Corp., New York, NY, IT-001–176 p).

## Cryo-EM

Purified BG505.SOSIP.664 trimers were mixed with threefold molar excess of QA013.2 Fab (fragment antigen binding) in 1X PBS at pH 7.4. The mixture was diluted with PBS to approximately 0.04 mg/mL final protein concentration and incubated for 45 min on ice. Lacey grids with a thin continuous carbon film (400 mesh) (Electron Microscopy Sciences, Hatfield, PA) were glow discharged (negative charge) under vacuum using 20–25 mA current for 30 s. A 3.0 μL aliquot of sample was applied

to these grids at 4°C and 100% humidity, blotted for 3.5 s and immediately plunge frozen in liquid ethane using a Vitrobot Mark IV specimen preparation unit (FEI Company, Hillsboro, OR).

Vitrified grids were imaged using an FEI Titan Krios operating at 300keV equipped with a Gatan K2 summit direct detector and a post-specimen energy filter. Micrographs were collected at ×105,000 magnification with a pixel size of 1.35Åper pixel in counting mode. Each image received a dose rate of ~8e$^-$/pixel/s with 200 ms exposure per frame and 50 frames per image collected. Estimated defocus ranged from 1.7 to 3.5 μm. Data were collected in two separate sessions with a total of 6475 micrographs using the automated data collection software Leginon (*Carragher et al., 2000*).

All data processing steps were performed within the Relion (version 3.0.7) software setup (*Scheres, 2012*). Frame alignment and dose-weighting were performed using MotionCor2 (*Zheng et al., 2017*), and CTF estimation was done using CTFFIND4 (*Rohou and Grigorieff, 2015*). Further processing was continued with Relion. A total of 1,082,742 particles were picked using the LOG-based automatic particle picking routine. The particles were initially extracted at 4x binning with a pixel size of 10.8 Å/pixel. The binned particle stack was subjected to two rounds of unsupervised 2D classification, out of which classes that clearly showed SOSIPs with three Fabs bound were selected. A total of 113,470 particles from these classes were used for 3D refinement. The previously published negative stain reconstruction of BG505.SOSIP.664 complexed to QA013.2 Fab (EMD-7471) was used as an initial model (*Williams et al., 2018*). The initial model was low-pass filtered to 60 Å and refinement was performed on the full particle stack with C3 symmetry imposed. Subsequent map sharpening and post processing in Relion resulted in a 4.186 Å structure using the 'gold-standard' FSC cutoff of 0.143. The unbinned particle stack was further subjected to CTF-refinement at per micrograph level which improved the map to its final resolution of 4.15 Å (*Figure 3—figure supplement 2*).

The atomic model of BG505.SOSIP.664 trimer (PDB ID: 5ACO) (*Lee et al., 2015*) was rigid body fitted into the 4.18 Å resolution map using UCSF Chimera (*Pettersen et al., 2004*). No further modification to the SOSIP model was performed as the atomic model agreed well with the map. In the cryo-EM map, the variable region of the Fab domain was well-resolved with separation in beta-sheets observable, but the constant domain of the Fab was only partially resolved. Thus, a QA013.2 Fab model consisting of only its variable heavy and light chains was generated using the SWISS model software (*Waterhouse et al., 2018*, p.). Five different models for QA013.2 Fab variable domain model were computationally generated using available PDB structure templates. All five models were comparable when superposed onto one another, except for the orientation of the variable CDRH3 loop. The Fab model generated by homology modeling with the crystal structure of an infant antibody ADI 19425 (PDB ID:6apc) (sequence identity of 71.18% and 0.97 coverage) was selected as the final model for further analysis.

The cryo-EM map was sharpened with BG505.SOSIP.664 atomic model (PDB ID:5ACO) input using the Autosharpen program (*Terwilliger et al., 2018*) in the Phenix software suite (*Liebschner et al., 2019*), in order to enhance features in the Fab density. The QA013.2 Fab model was then computationally docked into the map using 'Dock In Map' program in the Phenix software suite (*Liebschner et al., 2019*). The SOSIP+Fab model was then refined using Realspace Refinement in the Phenix package using reference model restraints from a high-resolution atomic model of BG505.SOSIP.664 (PDB ID:6mu6) (*Liebschner et al., 2019*). The data collection, refinement parameters and final model statistics are summarized in *Figure 3—source data 1*, and the cryo-EM map and coordinates are included in *Figure 3—source data 2*.

## Mutational antigenic profiling

Mutational antigenic profiling was performed as previously described (*Dingens et al., 2017*). Briefly, 0.5–1.0x10$^6$ TZM-bl infectious units of independent BG505.W6.C2 T332N mutant proviral DNA libraries (*Haddox et al., 2018*) were incubated with the QA013.2 bnAb at $\geq$ IC$_{95}$ concentration for one hour. One million SupT1.CCR5 cells were then infected with these mutant libraries in the presence of 100 μg/mL DEAE-dextran (VWR, Radnor, Pennsylvania, catalog #97061–684). In parallel to QA013.2 antibody selection, each mutant virus library was also infected into 1x10$^6$ SupT1.CCR5 cells without antibody selection to serve as the non-selected control. Non-integrated viral DNA was isolated from each pool of infected cells at 12 hr post infection using a Qiagen Miniprep kit. The *env* gene was amplified from each sample (selected and non-selected) using a barcoded subamplicon

sequencing approach that generates seven tiling subamplicons across *env* as previously described (*Haddox et al., 2018*). These pools of amplicons were then deep sequenced on an Illumina HiSeq using 250 bp paired-end reads.

Following deep sequencing, differential selection values were calculated as described in *Doud et al., 2017*. This included calculating the enrichment of each amino acid at individual sites relative to the wild-type by comparing the mutation counts of each residue in the antibody selected sample to the mock treated sample. Percent infectivity and correlation values between replicates are provided as *Figure 7—figure supplement 1*. We took the median values across all experimental replicates and visualized the differential selection data on logoplots rendered by dms_tools2 using weblogo (*Crooks et al., 2004*). Throughout this manuscript, we focused only on positively enriched mutations.

To validate the mutational antigenic profiling data, TZM-bl neutralization assays were performed using BG505.T332N pseudoviruses bearing single point mutants representing residues that were enriched for viral escape at individual sites across *Env*. TZM-bl neutralization assays were performed as described above in at least two independent assays, each performed in technical duplicate (see *Figure 7* – Neut Source Data 1). The fold change in $IC_{50}$ of each point mutant relative to BG505. T332N wild-type pseudovirus was calculated independently for each experiment and then averaged across all replicates.

## Figure generation

Portions of *Figures 1–2* and *4–6* were generated using Biorender.com. All figure panels were compiled using Adobe Illustrator.

## Data and code availability

Longitudinal antibody gene sequencing data of QA013's B cell repertoire can be found at BioProject SRA, accession PRJNA674442. The EM map and atomic coordinates for QA013.2 complexed to BG505.SOSIP.664 are deposited under accession codes EMD-24195 and PDB 7N65.

The computationally inferred antibody variable region sequences, chimeras, and mutant antibody chains produced in this study have not been deposited into GenBank because computationally-generated sequences are not accepted, but they are available in *Supplementary files 1* and *3*. Mature QA013.2 antibody chain GenBank accession numbers are MH003558.1 (VH) and MH003564.1 (VL). GenBank accession numbers for HIV Env variants used in this study can be found in the Key Resources Table above.

The open-source software used in this study is publicly available on GitHub: *partis* (https://github.com/psathyrella/partis; *Ralph, 2021a*), pruning of clonal family sequences (https://github.com/matsengrp/cft/blob/master/bin/prune.py; *Harkins, 2021a*), *linearham* (https://github.com/matsengrp/linearham; *Jasti, 2021*), and Blast validation (https://github.com/matsengrp/cft/blob/master/bin/blast.py; *Harkins, 2021b*). The specific commands and run logs for the lineage analysis can be found on Zenodo. Open-source software to analyze mutational antigenic profiling datasets is available at https://jbloomlab.github.io/dms_tools2/ (*Bloom, 2021*).

## Acknowledgements

We thank all of the participants, staff, Scott McClelland, and Ludo Lavreys for their continued involvement and contributions to the Mombasa cohort. We acknowledge Vladimir Vigdorovich and D Noah Sather for their technical assistance with NGS library preparation and sequence processing. This work was supported by NIH grants R01 AI140891 to JDB, R01 AI146028, U19 AI117891, and U19 AI128914 FAM, R01 AI140868 to KKL, and R01 AI138709 to JO JDB is an investigator of the Howard Hughes Medical Institute. The research of Frederick Matsen was supported in part by a Faculty Scholar grant from the Howard Hughes Medical Institute and the Simons Foundation.

# Additional information

## Competing interests

Julie M Overbaugh: Reviewing editor, *eLife*. The other authors declare that no competing interests exist.

## Funding

| Funder | Grant reference number | Author |
| --- | --- | --- |
| National Institutes of Health | R01 AI140891 | Jesse D Bloom |
| National Institutes of Health | R01 AI146028 | Frederick A Matsen IV |
| National Institutes of Health | U19 AI117891 | Frederick A Matsen IV |
| National Institutes of Health | U19 AI128914 | Frederick A Matsen IV |
| National Institutes of Health | R01 AI140868 | Kelly K Lee |
| National Institutes of Health | R01 AI138709 | Julie M Overbaugh |

The funders had no role in study design, data collection and interpretation, or the decision to submit the work for publication.

## Author contributions

Mackenzie M Shipley, Data curation, Formal analysis, Investigation, Visualization, Writing - original draft, Project administration, Writing - review and editing; Vidya Mangala Prasad, Data curation, Formal analysis, Investigation, Visualization, Writing - review and editing; Laura E Doepker, Data curation, Methodology, Writing - review and editing; Adam Dingens, Data curation, Formal analysis, Methodology, Writing - review and editing; Duncan K Ralph, Software, Formal analysis, Methodology, Writing - review and editing; Elias Harkins, Software, Formal analysis, Visualization, Methodology; Amrit Dhar, Software, Formal analysis, Methodology; Dana Arenz, Data curation; Vrasha Chohan, Haidyn Weight, Data curation, Validation; Kishor Mandaliya, Resources, Supervision, Project administration; Jesse D Bloom, Resources, Software, Funding acquisition; Frederick A Matsen IV, Resources, Software, Supervision, Funding acquisition, Methodology, Project administration, Writing - review and editing; Kelly K Lee, Resources, Supervision, Funding acquisition, Project administration, Writing - review and editing; Julie M Overbaugh, Conceptualization, Resources, Supervision, Funding acquisition, Project administration, Writing - review and editing

## Author ORCIDs

Mackenzie M Shipley (ID) https://orcid.org/0000-0002-7436-5622
Laura E Doepker (ID) http://orcid.org/0000-0003-4514-5003
Adam Dingens (ID) https://orcid.org/0000-0001-9603-9409
Jesse D Bloom (ID) http://orcid.org/0000-0003-1267-3408
Julie M Overbaugh (ID) https://orcid.org/0000-0002-0239-9444

## Ethics

Human subjects: This study (Clinical Trial Management System Number RG1000880) was approved by members of the ethical review committees (file number 7776) at the University of Nairobi, the Fred Hutchinson Cancer Research Center, and the University of Washington. Study participants provided written informed consent prior to enrollment.

## Decision letter and Author response

Decision letter https://doi.org/10.7554/eLife.68110.sa1
Author response https://doi.org/10.7554/eLife.68110.sa2

# Additional files

## Supplementary files

• Supplementary file 1. Fasta file of all computationally-inferred QA013.2 lineage members. Sequence file includes inferred naive BCRs, computationally inferred lineage intermediates, and mature sequences for both VH and VL lineages.

• Supplementary file 2. Fasta file of sampled NGS sequences with high percent identity to computationally inferred lineage members. Sequence file includes sampled NGS sequences with the highest nucleotide identity to all QA013.2 inferred lineage members and mature sequences.

• Supplementary file 3. Fasta file of all QA013.2 VH and VL chimeras and mutant variants. Sequence file includes all VH and VL chimeras (presented in *Figures 4* and *6*), and VH mutants (presented in *Figure 5*) generated to interrogate QA013.2 functional evolution in this study.

• Transparent reporting form

## Data availability

Sequencing data have been deposited at BioProject SRA, accession PRJNA674442. The EM map and atomic coordinates for QA013.2 complexed to BG505.SOSIP.664 are deposited under accession codes EMD-24195 and PDB 7N65. Source data have been provided for Figures 2-8.The specific commands and run logs for the lineage analysis can be found on Zenodo.

The following datasets were generated:

| Author(s) | Year | Dataset title | Dataset URL | Database and Identifier |
|---|---|---|---|---|
| Shipley MM, Doepker LE, Arenz DA, Ralph D, Matsen FA, Overbaugh JM | 2020 | Subject QA013 antibody sequencing | https://www.ncbi.nlm.nih.gov/bioproject/?term=PRJNA674442 | NCBI BioProject, PRJNA674442 |
| Shipley MM, Doepker LE, Dingens A, Ralph DK, Harkins E, Dhar A, Arenz D, Chohan V, Weight H, Mandaliya K, Bloom JD, Matsen FA, Lee KK, Overbaugh JM, Mangala PV | 2021 | Functional development of a V3/glycan-specific broadly neutralizing antibody isolated from a case of HIV superinfection. | https://doi.org/10.2210/pdb7n65/pdb | Worldwide Protein Data Bank, 10.2210/pdb7n65/pdb |
| Shipley MM, Mangala Prasad V, Doepker LE, Dingens A, Ralph DK, Harkins E, Dhar A, Chohan V, Weight H, Mandaliya K, Bloom JD, Matsen FA, Lee KK, Overbaugh JM | 2021 | Complex structure of HIV superinfection Fab QA013.2 and BG505.SOSIP.664 | https://www.rcsb.org/structure/7N65 | RCSB Protein Data Bank, 7N65 |
| Shipley MM, Mangala Prasad V, Ralph DK, Harkins E, Dhar A, Matsen FA, Lee KK, Overbaugh JM | 2021 | Functional development of a V3-specific broadly neutralizing antibody isolated from a case of HIV superinfection | https://doi.org/10.5281/zenodo.4574672 | Zenodo, 10.5281/zenodo.4574672 |

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
