## [Decision Letter]

**Acceptance summary:**

Your manuscript adds to the understanding of broadly neutralizing antibodies against HIV by describing the development and maturation of a broadly neutralizing antibody derived from a donor who was infected with two different subclasses of HIV. The study highlights the differences between this monoclonal antibody and others directed at the same site on HIV Envelope, suggesting multiple ways the immune system has to evolve similar antibodies to combat this virus infection.

**Decision letter after peer review:**

Thank you for submitting your article "Functional development of a V3/glycan-specific broadly neutralizing antibody isolated from a case of HIV superinfection" for consideration by *eLife*. Your article has been reviewed by 3 peer reviewers, and the evaluation has been overseen by a Reviewing Editor and Cynthia Wolberger as the Senior Editor. The following individual involved in review of your submission has agreed to reveal their identity: Suzana Zolla-Pazner (Reviewer #1).

Essential Revisions:

1) The study covers antibody deep sequencing data and function using sequential specimens from pre-HIV infection to 765 days post-initial infection (dpii). The mAb QA013.2 was isolated at 2282 dpii. More than 70% of the somatic mutations occurred between 764 and 2282 dpii, a period not covered by the study. Were samples available from days 764-2282, and if so, why were these specimens not deep sequenced? To fully understand the driving force behind the intermediate forms and the mature form of the monoclonal antibody, it would be optimal to study the intermediates between days 764 and 2282 if this is possible.

2) In terms of novelty, the appearance of this HIV broadly neutralizing lineage and its development prior to HIV infection is of substantial interest to the field. Unfortunately, this important result is not noted in the abstract, nor substantially analyzed. The extensive interaction of this antibody with high mannose glycans at residues N301 and N332 suggest initiation of the lineage by a glycosylated antigen, and it would be helpful for the authors to focus a bit more experimentally (and in the discussion) on this novel aspect of the lineage. In particular, the Abstract should better reflect the novel aspects of the QA013.2 bnAb compared to other V3-glycan bnAbs. This monoclonal Ab (a) relies less on the CDRH3 loop and more on framework mutations for development of neutralization breadth and affinity; (b) the lineage may have been influenced by both the initial and superinfecting viruses in the donor; and (c) virus escape mutations from QA013.2 occur throughout the V3 loop of Env and in V1 unlike other V3-glycan bnAbs that have more focused escape signatures.

3) While conclusions are generally well supported, the statement "QA013.2 requires residues spanning FWRH1-CDRH1 to attain breadth" is not well substantiated. The authors use chimeras to show that substitution of SHM in the FWRH1-CDRH1 are important for breadth, but the analysis of FWRH1 residues is lacking: in specific, none of the chimeras specifically revert FWRH1 segment, while leaving the rest of the antibody mature; thus the chimeric data does not specifically implicate FWRH1 in function. In terms of individual mutations, the first residue with a substantial impact on breadth is E26G, which is part of the CDRH1. As it's already well established that CDRH1 can impact antibody function, the authors should either downplay or remove parts of the paper concluding FWRH1 residues are important or provide specific experimental data showing FWRH1 residues do substantially impact function (the minor impact at residue 24 and 25 on Tier 1 isolate SF162 seems too subtle to support a highlight that FWRH1 residues are critical.

4) Adding some additional information to the paper would be helpful. For the initial clade D variant, the superinfecting clade A variant, and all of the other HIV-1 Env proteins and pseudoviruses that were used, the authors could include a table or figure that shows whether each contains N332, N301, and GDIR. It would also be helpful to use a wider spectrum of colors in the heatmaps showing neutralizing IC50s to highlight differences more easily. Regarding the N301 glycan, mutant pseudoviruses representing the antigenic profiling escape mutations at N301/T303 tested could be tested for neutralization resistance in the TZM-bl assay, to confirm the dependence of QA013.2 on this glycan in addition to N332.

5) Please show the number of lineage transcripts identified at each time point in Figure 1A.

6) Figure 1BC – It would be helpful, if possible, to show temporal development of intermediates. It seems like a lot of development is occurring before infection. Does this development enhance antibody affinity to high mannose N-linked glycans?

7) It would be helpful to clarify the neutralization breadth of the mature version of the antibody. In particular, it would be helpful to characterize on a decent panel of viral isolates (50-100). The only data I could find was in the prior Williams 2018 paper, on a panel of ~20 strains, where no Tier 2 clade A strains could be neutralized (yet the mature antibody clearly binds to BG505 from clade A).

8) Table 2 – I like ending with a summary figure – but it would helpful to add location of framework 1 mutations that impact function (if these are indeed identified).

9) It would be helpful in the discussion to compare with other HIV-1 neutralizing antibodies, which have been suggested to be initiated against non-HIV antigens, such as the fascinating microbiome initiated gp41-directed antibodies described by Haynes and colleagues.

10) Please make the list summarizing literature on antibodies lineage reconstruction (lines 59-64) more comprehensive. I can see a number of Immunity papers should be listed including the Kong et al. study on VRC01 antibodies (Immunity 2016), the Landais et al. study on V2-apex antibodies, the Krebs et al. study on MPER antibodies (Immunity 2019) and the Umotoy et al. study on VRC01 antibodies (Immunity 2019) should be added.

11) 4) The term "V3/glycan-specific" bnAbs should be used throughout rather that V3-specific Abs.

---

## [Author Response]

Essential Revisions:1) The study covers antibody deep sequencing data and function using sequential specimens from pre-HIV infection to 765 days post-initial infection (dpii). The mAb QA013.2 was isolated at 2282 dpii. More than 70% of the somatic mutations occurred between 764 and 2282 dpii, a period not covered by the study. Were samples available from days 764-2282, and if so, why were these specimens not deep sequenced? To fully understand the driving force behind the intermediate forms and the mature form of the monoclonal antibody, it would be optimal to study the intermediates between days 764 and 2282 if this is possible.

We appreciate the reviewers’ suggestions, however given that there are limited samples available from this subject, we are trying to balance how they are used. We considered that sequencing an additional sample(s) from later timepoints would likely be incremental, and a bit peripheral to the major point of this paper. While the sequencing data we present in this study was not sufficient to completely resolve the evolution of this antibody lineage, we feel that the main findings of the study are robust. In particular, here we aimed to combine the sequencing data with cryo-EM and functional studies to investigate the structure-function relationships of the V3/glycan-specific bnAb QA013.2. The combination of these techniques allowed us to define the key functional and structural features that resulted in the development and breadth of the QA013.2 bnAb. But we recognize that these data were not sufficient to precisely delineate the evolutionary pathway of bnAb development in the individual, which we are careful not to claim.

2) In terms of novelty, the appearance of this HIV broadly neutralizing lineage and its development prior to HIV infection is of substantial interest to the field. Unfortunately, this important result is not noted in the abstract, nor substantially analyzed. The extensive interaction of this antibody with high mannose glycans at residues N301 and N332 suggest initiation of the lineage by a glycosylated antigen, and it would be helpful for the authors to focus a bit more experimentally (and in the discussion) on this novel aspect of the lineage. In particular, the Abstract should better reflect the novel aspects of the QA013.2 bnAb compared to other V3-glycan bnAbs. This monoclonal Ab (a) relies less on the CDRH3 loop and more on framework mutations for development of neutralization breadth and affinity; (b) the lineage may have been influenced by both the initial and superinfecting viruses in the donor; and (c) virus escape mutations from QA013.2 occur throughout the V3 loop of Env and in V1 unlike other V3-glycan bnAbs that have more focused escape signatures.

We appreciate the Reviewers’ comments regarding the novelty of this particular superinfection case. We have revised the Abstract to more clearly express our major findings (lines 34-42), taking into account the word restrictions (150) and balancing these two aspects.

“BnAb QA013.2 bound initial and superinfecting viral Env, despite its probable naïve progenitor only recognizing the superinfecting strain, suggesting both viruses influenced this lineage. A 4.15 Å cryo-EM structure of QA013.2 bound to native-like trimer showed recognition of V3 signatures (N301/N332 and GDIR). QA013.2 relies less on CDRH3 and more on framework and CDRH1 for affinity and breadth compared to other V3/glycan-specific bnAbs. Antigenic profiling revealed that viral escape was achieved by changes in the structurally-defined epitope and by mutations in V1. These results highlight shared and novel properties of QA013.2 relative to other V3/glycan-specific bnAbs in the setting of sequential, diverse antigens.”

In addition, as discussed below in response to point #4 below, we did further explore the role of the N301 and N332 glycans.

The Reviewers also mention the novelty of the appearance of this lineage prior to infection, which we agree is an interesting finding. We now note this in the discussion (lines 592-598), but did not want to overemphasize these data since we did not explore this further in this study because it was not the focus.

3) While conclusions are generally well supported, the statement "QA013.2 requires residues spanning FWRH1-CDRH1 to attain breadth" is not well substantiated. The authors use chimeras to show that substitution of SHM in the FWRH1-CDRH1 are important for breadth, but the analysis of FWRH1 residues is lacking: in specific, none of the chimeras specifically revert FWRH1 segment, while leaving the rest of the antibody mature; thus the chimeric data does not specifically implicate FWRH1 in function. In terms of individual mutations, the first residue with a substantial impact on breadth is E26G, which is part of the CDRH1. As it's already well established that CDRH1 can impact antibody function, the authors should either downplay or remove parts of the paper concluding FWRH1 residues are important or provide specific experimental data showing FWRH1 residues do substantially impact function (the minor impact at residue 24 and 25 on Tier 1 isolate SF162 seems too subtle to support a highlight that FWRH1 residues are critical.

We appreciate the suggestion made by the Reviewers. To address this concern directly, we synthesized a heavy chain variant of the mature VH that had only FWRH1 reverted back to the probable naïve BCR (∆FWRH1) as suggested by the reviewer. When we tested the neutralization capacity of this VH variant paired with the mature VL, we found that the only heterologous virus that was neutralized was QC406.F3, which was weakly neutralized at the highest antibody concentration tested (IC_50_ = 44 µg/mL). Interestingly, the autologous clade A virus IC_50_ for this new chimeric antibody was identical to the ∆CDRH1 chimeric antibody we synthesized previously, which had only the six consecutive residues spanning FWRH1 and CDRH1 (∆CDR1_VH_) reverted back to the inferred naïve VH sequence (IC_50_ = 17 µg/mL). We have included these data into Figure 4 and added the replicate data to the figure’s source dataset. Given these findings support our original conclusions that the V3/glycan-specific bnAb QA013.2 requires residues spanning FWRH1-CDRH1 to attain neutralization breadth and function, we left the discission of this aspect unchanged.

4) Adding some additional information to the paper would be helpful. For the initial clade D variant, the superinfecting clade A variant, and all of the other HIV-1 Env proteins and pseudoviruses that were used, the authors could include a table or figure that shows whether each contains N332, N301, and GDIR. It would also be helpful to use a wider spectrum of colors in the heatmaps showing neutralizing IC50s to highlight differences more easily. Regarding the N301 glycan, mutant pseudoviruses representing the antigenic profiling escape mutations at N301/T303 tested could be tested for neutralization resistance in the TZM-bl assay, to confirm the dependence of QA013.2 on this glycan in addition to N332.

These are excellent suggestions by the Reviewers and we have incorporated these recommendations into the manuscript. More specifically, we have done the following:

– Expanded the color scheme used to delineate neutralization potency in all neutralization tables.

– Added in relevant characteristics for each pseudovirus tested, including the presence of N301, N332, GDIR, and the residue content of the final amino acids in V1. This has been added as a new table (Table 2).

– Tested QA013.2 bnAb against BG505 T332N N301K mutant virus. These new data are now shown as a part of new Table 2.

5) Please show the number of lineage transcripts identified at each time point in Figure 1A.

We have added this information to Figure 1 panel A.

6) Figure 1BC – It would be helpful, if possible, to show temporal development of intermediates. It seems like a lot of development is occurring before infection. Does this development enhance antibody affinity to high mannose N-linked glycans?

There are two points raised here by the Reviewers. The first is related to the above inquiry (#5) regarding the number of transcripts that were identified at each longitudinal sequencing timepoint. We have added this information to Figure 1A, which sheds light on the temporal development that is occurring in this lineage. However, we’d like to point out that deep sequencing B cell repertoires is inconsistent, with coverage that is rarely ever “complete” due to fluctuations in the B cell repertoire over time (Horns et al., 2019; Laserson et al., 2014) and the fact that PBMC samples only represent a small fraction of the whole blood volume. Therefore, it is possible that we may have missed sampling lineage transcripts that were circulating in the blood of this subject at each longitudinal timepoint.

The second point regards the role of glycans in this process, which we felt was difficult to address directly based on new experimental data in Table 2, which demonstrate that the N301 glycan, in addition to the glycan at N332, are both required to achieve neutralization by QA013.2. This may suggest that affinity and/or avidity to high mannose N-linked glycans plays a role in this antibody’s lineage development. However, we note that this result would make it challenging to use glycan arrays or biolayer interferometry (BLI) to accurately address this point. That is because if both glycans are needed together to achieve bnAb binding and neutralization, as our new data suggests, then it is unlikely that we would observe any signal of binding to these individual glycan moieties when presented in a glycan array or BLI format. Moreover, we did explore this idea and the turnaround time for commercial glycan array testing is considerable (4-6 weeks). We do not want to delay the manuscript any further, given the limitations of this approach noted above, but if the Reviewers feel that investigating this question is critical for our manuscript then we will be willing to delay the paper to oblige.

7) It would be helpful to clarify the neutralization breadth of the mature version of the antibody. In particular, it would be helpful to characterize on a decent panel of viral isolates (50-100). The only data I could find was in the prior Williams 2018 paper, on a panel of ~20 strains, where no Tier 2 clade A strains could be neutralized (yet the mature antibody clearly binds to BG505 from clade A).

We appreciate the Reviewer’s request for an expanded panel; we would like to point out that our prior publication (Williams et al., Cell Reports 2018) tested 31 total virus strains, including 12 from the Global Panel, which is a panel specifically designed to use a more focused, well-defined panel for these types of studies. To address the Reviewer’s request for a 50-virus panel, we have included an additional 19 pseudoviruses from both clade A and clade D – the two clades that are most relevant to subject QA013 and this bnAb. Importantly, all of the included strains were neutralized by V3/glycan-specific bnAbs PGT121 and/or PGT128 as shown previously (Mabuka et al., AIDS 2013 and Goo et al., J. Virology 2012). This expanded panel has been included as a new figure (Table 2).

8) Table 2 – I like ending with a summary figure – but it would helpful to add location of framework 1 mutations that impact function (if these are indeed identified).

As we mention in our response to inquiry #3 above, we do in fact find that reversion of the entire framework 1 region of the mature VH alone results in complete loss of neutralization function across the three clades of pseudoviruses tested. Our summary figure already highlights FWRH1 on the cryo-EM structure as a region of interest for QA013.2 neutralization function and breadth.

9) It would be helpful in the discussion to compare with other HIV-1 neutralizing antibodies, which have been suggested to be initiated against non-HIV antigens, such as the fascinating microbiome initiated gp41-directed antibodies described by Haynes and colleagues.

We appreciate this point and have added several sentences to the Discussion (lines 592-598) mentioning how the presence of members of this bnAb lineage pre-HIV infection is reminiscent of gp41-specific antibodies that were found to be initiated by commensal bacteria present in the gut. However, as we mention above in our response to point #2, we did not want to overstate these data as we did not explore this finding further for this specific study.

“From the deep sequencing data we observed the presence of early bnAb lineage members circulating in the blood pre-HIV infection. This suggests that this antibody lineage may have been stimulated by an antigen other than HIV, as has been suggested for some gp41-specific antibodies, which recognize commensal bacteria in the gut (Liao et al., 2011; Trama et al., 2014; Williams et al., 2015). Given the dependence of QA013.2 bnAb on the two glycans at N301 and N332 in combination, we speculate that this bnAb lineage may have been initiated by a glycosylated antigen prior to HIV infection.”

10) Please make the list summarizing literature on antibodies lineage reconstruction (lines 59-64) more comprehensive. I can see a number of Immunity papers should be listed including the Kong et al. study on VRC01 antibodies (Immunity 2016), the Landais et al. study on V2-apex antibodies, the Krebs et al. study on MPER antibodies (Immunity 2019) and the Umotoy et al. study on VRC01 antibodies (Immunity 2019) should be added.

We thank the Reviewers for pointing out our incomplete list of references for antibody lineage reconstruction. We have added the mentioned papers to the relevant portion of the Introduction section (lines 60-66).

11) 4) The term "V3/glycan-specific" bnAbs should be used throughout rather that V3-specific Abs.

This term has been updated throughout the manuscript text.References

Horns F, Vollmers C, Dekker CL, Quake SR. 2019. Signatures of selection in the human antibody repertoire: Selective sweeps, competing subclones, and neutral drift. *Proc Natl Acad Sci USA* 116:1261–1266. doi:10.1073/pnas.1814213116

Laserson U, Vigneault F, Gadala-Maria D, Yaari G, Uduman M, Vander Heiden JA, Kelton W, Taek Jung S, Liu Y, Laserson J, Chari R, Lee J-H, Bachelet I, Hickey B, Lieberman-Aiden E, Hanczaruk B, Simen BB, Egholm M, Koller D, Georgiou G, Kleinstein SH, Church GM. 2014. High-resolution antibody dynamics of vaccine-induced immune responses. *Proc Natl Acad Sci USA* 111:4928–4933. doi:10.1073/pnas.1323862111

Liao H-X, Chen X, Munshaw S, Zhang R, Marshall DJ, Vandergrift N, Whitesides JF, Lu X, Yu J-S, Hwang K-K, Gao F, Markowitz M, Heath SL, Bar KJ, Goepfert PA, Montefiori DC, Shaw GC, Alam SM, Margolis DM, Denny TN, Boyd SD, Marshal E, Egholm M, Simen BB, Hanczaruk B, Fire AZ, Voss G, Kelsoe G, Tomaras GD, Moody MA, Kepler TB, Haynes BF. 2011. Initial antibodies binding to HIV-1 gp41 in acutely infected subjects are polyreactive and highly mutated. *Journal of Experimental Medicine* 208:2237–2249. doi:10.1084/jem.20110363

Trama AM, Moody MA, Alam SM, Jaeger FH, Lockwood B, Parks R, Lloyd KE, Stolarchuk C, Scearce R, Foulger A, Marshall DJ, Whitesides JF, Jeffries TL, Wiehe K, Morris L, Lambson B, Soderberg K, Hwang K-K, Tomaras GD, Vandergrift N, Jackson KJL, Roskin KM, Boyd SD, Kepler TB, Liao H-X, Haynes BF. 2014. HIV-1 Envelope gp41 Antibodies Can Originate from Terminal Ileum B Cells that Share Cross-Reactivity with Commensal Bacteria. *Cell Host & Microbe* 16:215–226. doi:10.1016/j.chom.2014.07.003

Williams WB, Liao H-X, Moody MA, Kepler TB, Alam SM, Gao F, Wiehe K, Trama AM, Jones K, Zhang R, Song H, Marshall DJ, Whitesides JF, Sawatzki K, Hua A, Liu P, Tay MZ, Seaton KE, Shen X, Foulger A, Lloyd KE, Parks R, Pollara J, Ferrari G, Yu J-S, Vandergrift N, Montefiori DC, Sobieszczyk ME, Hammer S, Karuna S, Gilbert P, Grove D, Grunenberg N, McElrath MJ, Mascola JR, Koup RA, Corey L, Nabel GJ, Morgan C, Churchyard G, Maenza J, Keefer M, Graham BS, Baden LR, Tomaras GD, Haynes BF. 2015. Diversion of HIV-1 vaccine-induced immunity by gp41-microbiota cross-reactive antibodies. *Science* 349:aab1253–aab1253. doi:10.1126/science.aab1253